# Retraining-Free Merging of Sparse MoE via Hierarchical Clustering

**I-Chun Chen** [1]  **Hsu-Shen Liu** [1]  **Wei-Fang Sun** [2]  **Chen-Hao Chao** [3]  **Yen-Chang Hsu** [4]  **Chun-Yi Lee** [5]

## Abstract

Sparse Mixture-of-Experts (SMoE) models represent a significant advancement in large language model (LLM) development through their efficient parameter utilization. These models achieve substantial performance improvements at reduced inference costs. However, the deployment of SMoE models faces constraints from extensive memory requirements of expert components in resource-limited environments. To address these limitations, this paper introduces Hierarchical Clustering for Sparsely activated Mixture of Experts (HC-SMoE), a task-agnostic expert merging framework for parameter reduction without retraining. HC-SMoE introduces a novel hierarchical clustering approach based on expert outputs to ensure merging robustness independent of routing decisions. The proposed output-based clustering method enables effective capture of functional relationships between experts for large-scale architectures. We provide theoretical analysis and comprehensive evaluations across multiple zero-shot language tasks to demonstrate HC-SMoE's effectiveness in state-of-the-art models including Qwen and Mixtral. The experimental results validate HC-SMoE's superior performance and practical applicability for real-world deployments. Our implementation is available at https://github.com/wazenmai/HC-SMoE.

## 1. Introduction

Transformer-based architectures in natural language processing (NLP) have demonstrated significant performance improvements across various tasks with exponential parameter growth (Chowdhery et al., 2022; OpenAI et al., 2024;

---

[1]Department of Computer Science, National Tsing Hua University, Taiwan [2]NVIDIA AI Technology Center (NVAITC) [3]Department of Computer Science, University of Toronto, Canada [4]Samsung Research America [5]Department of Computer Science and Information Engineering, National Taiwan University, Taiwan. Correspondence to: Chun-Yi Lee <cylee@csie.ntu.edu.tw>.

*Proceedings of the $42^{nd}$ International Conference on Machine Learning*, Vancouver, Canada. PMLR 267, 2025. Copyright 2025 by the author(s).

Team et al., 2024). This increase in size creates substantial challenges for real-world deployment due to heightened inference latency and computational demands (Bommasani et al., 2022). Sparsely activated Mixture of Experts (SMoE) models offer a promising solution through their sparse activation mechanism, where only selected model parameters ('*experts*') activate per input token. This architecture enables extensive parametric capacity without proportional computational costs during inference (Shazeer et al., 2017; Fedus et al., 2022). However, the total size of SMoE architectures presents significant memory constraints, which positions parameter reduction as a critical research priority. Recent studies have revealed important insights about expert redundancy. Liu et al. (2023) identified high representational similarities among experts and proposed methods to enhance expert diversity. Lu et al. (2024) provided additional empirical evidence to support these findings. These studies demonstrate substantial parameter redundancy in current SMoEs and highlight opportunities for optimization.

Several approaches have emerged to address redundant parameters in SMoE models. Early research introduced task-specific expert pruning (Chen et al., 2022), which progressively eliminates non-essential experts and produces a single-expert dense model for specific downstream tasks. However, such approaches often necessitate extensive fine-tuning to maintain performance levels. Recent studies have advanced toward retraining-free expert pruning methods (Lu et al., 2024; He et al., 2024). Lu et al. (2024) proposed an approach to trim experts based on output loss comparison with the original model. He et al. (2024) developed a more scalable solution through routing score-based pruning. However, the complete removal of experts and their parameters can cause irreversible loss of learned representations. An alternative research direction explores expert merging rather than pruning. Li et al. (2024) introduced a method to consolidate information from significant experts. Nevertheless, our experimental results presented in Section 4 reveal limitations in this methodology's task-agnostic generalizability.

In response to these challenges, this paper introduces the **H**ierarchical **C**lustering for **S**parsely Activated **M**ixture **of** **E**xperts (**HC-SMoE**), a *retraining-free*, *scalable*, and *task-agnostic* framework. HC-SMoE reduces SMoE model parameters through hierarchical clustering based on expert outputs and frequency-weighted merging. The hierarchi-

cal clustering methodology provides two significant advantages. First, the approach advances beyond the single-pass grouping method from Li et al. (2024) through iterative comparisons, which maintains superior inter-cluster diversity and intra-cluster similarity. Second, HC-SMoE utilizes expert outputs rather than router logits as its similarity metric, which enhances generalizability beyond dataset-specific patterns. For a fair comparison with previous work, we follow standard evaluation protocols with clustering and merging on the C4 dataset (Raffel et al., 2020) and evaluate accuracy across eight zero-shot language tasks (Lu et al., 2024). Our extensive evaluations in the supplementary material further demonstrate HC-SMoE's effectiveness across diverse dataset domains and tasks. Our results in Fig.1 illustrates HC-SMoE's achievement of comparable performance to the original Qwen model, with improvements of 6.95% and 2.14% over the strongest baseline in 8B and 11B parameter configurations. Section 4 further validates HC-SMoE's superior performance across all baselines on Mixtral $8 \times 7$B. The primary contributions of this study are summarized as:

- This work proposes HC-SMoE, a promising retraining-free and task-agnostic merging strategy with efficient scaling characteristics for different numbers of experts.

- We propose expert outputs as an effective similarity metric for clustering, which offers advantages over traditional router logits or weights from prior approaches.

- Our analysis substantiates the importance of clustering quality for merging effectiveness. The proposed hierarchical clustering method produces theoretically guaranteed and empirically validated expert groupings.

- Our extensive experiments demonstrate HC-SMoE's consistent superior performance across diverse benchmarks and its effectiveness on multiple SMoE models.

## 2. Background and Related Works

### 2.1. Sparsely Activated Mixture-of-Experts (SMoE)

The SMoE model comprises multiple SMoE layers, each of which contains a set of expert neural networks and a router network. Consider an input token $x$, a set of expert neural networks $\{E_1, E_2, ..., E_n\}$, and a router network $R$. The output $y$ of an SMoE layer is computed as a weighted sum of the expert network outputs, which can be expressed as:

$$y = \sum_{i=1}^{n} P_i(x) \cdot E_i(x), \tag{1}$$

$$E(x) = (\sigma(xW_{\text{gate}}) \odot (xW_{\text{up}}))W_{\text{down}}, \tag{2}$$

where $P_i(x)$ represents the $i^{th}$ expert routing score from $R$, and $E_i(x)$ denotes the $i^{th}$ expert network output. This architecture extends to recent models like Qwen (Team,

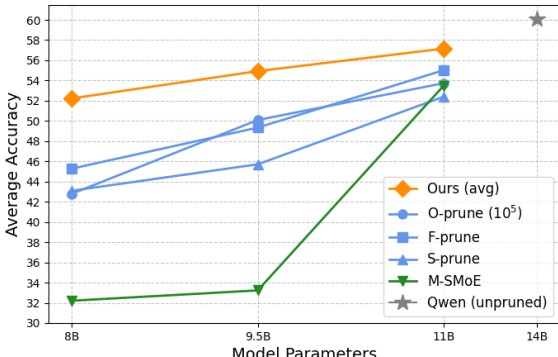

*Figure 1.* Effectiveness of expert parameter reduction approaches on Qwen1.5-MoE-A2.7B-Chat (Team, 2024). Average accuracy across 8 LM-Harness benchmarks demonstrates HC-SMoE's superior performance over existing retraining-free pruning and merging baselines at 25%, 37.5%, and 50% expert parameter reduction rates. ★ indicates the original unpruned Qwen model performance.

2024) and Mixtral (Jiang et al., 2024), which adopt the LLaMA (Touvron et al., 2023) structure. The feed-forward network (FFN) in each expert implements three linear layers as shown in Eq. (2), where element-wise multiplication $\odot$ operates with weight matrices $W_{\text{up}}, W_{\text{gate}} \in \mathbb{R}^{d_h \times d_m}$, $W_{\text{down}} \in \mathbb{R}^{d_m \times d_h}$, and Sigmoid Linear Unit (SiLU) activation function $\sigma$ (Elfwing et al., 2018). The routing implementation employs an efficient top-$k$ strategy (Shazeer et al., 2017; Fedus et al., 2022) to select experts with the highest logits from linear input transformation. A subsequent softmax operation on these $k$ largest logits enables sparse expert activation, which reduces computational overhead. This selective mechanism is formulated as follows:

$$P(x) = \text{softmax}(\text{topK}(R(x))) = \text{softmax}(\text{topK}(xW_R)), \tag{3}$$

where $R(x)$ represents routing-logits and $W_R$ denotes the learnable parameter matrix. This sparsely activated architecture enables efficient scaling with preserved performance through selective computation. In turn, this mechanism allows the SMoE model to optimize computational efficiency and task performance through focused expert utilization.

### 2.2. Expert Pruning and Merging

This section reviews existing methods for expert reduction in SMoE architectures, summarized in Table 1. Recent research has focused on various pruning strategies. Chen et al. (2022) proposed Task-Specific Expert Pruning (TSEP), which reduces active experts through iterative fine-tuning for specific downstream tasks. Although effective, this approach requires extensive computational resources and time for fine-tuning, which limits its applicability to large-scale models. Lu et al. (2024) introduced a method, which we refer to as *O-prune* in this study, for retraining-free and task-agnostic expert reduction in zero-shot settings. The method

*Table 1.* A Comparison of different approaches for reducing the number of experts in SMoE. We evaluate approaches based on their retraining-free nature, task-agnostic applicability, scalability, and strategies. Our method, HC-SMoE, is compared with TSEP (Chen et al., 2022), O-prune (Lu et al., 2024), S-prune (He et al., 2024), F-prune, and M-SMoE (Li et al., 2024). Please note that F-prune is detailed in Section 4.

| Method | Retraining-free | Task-agnostic | Scalable | Strategy |
|--------|:---------------:|:-------------:|:--------:|----------|
| TSEP | ✗ | ✗ | ✓ | Pruning |
| O-prune | ✓ | ✓ | ✗ | Pruning |
| S-prune | ✓ | ✓ | ✓ | Pruning |
| F-prune | ✓ | ✓ | ✓ | Pruning |
| M-SMoE | ✗ | ✗ | ✓ | Merging |
| HC-SMoE | ✓ | ✓ | ✓ | Merging |

determines expert retention counts per layer and evaluates all possible expert combinations to select configurations that minimize output deviation from the original model. However, this approach discards potential knowledge from pruned experts. Moreover, its computational requirements become prohibitive for large expert counts. For example, a 50% reduction in Qwen's 60 experts requires evaluation of approximately $C(60, 30) \approx 10^{18}$ combinations per layer. He et al. (2024) proposed an efficient expert trimming technique, denoted as *S-prune*, based on router scores. This method accumulates global router-scores $P(x)$ and retains top-scoring experts, which offer enhanced flexibility over *O-prune* by allowing variable expert retention across layers.

Model merging techniques have emerged as a promising approach to combine the strengths of multiple models. ZipIt (Stoica et al., 2024) introduces a model merging technique that allows models with the same architecture but trained on different tasks to be merged without retraining. It utilizes pairwise feature correlation to merge features both within a single model and across different models, and provides flexibility in choosing correlated features. Since expert merging can be considered a multi-model merging problem, we extend ZipIt to this context. However, its extensive feature correlation computation makes it time-consuming and less effective for expert merging scenarios. M-SMoE (Li et al., 2024) proposes a three-step pipeline for expert merging in SMoE models. It first selects dominant experts based on activation frequency to decide which experts to retain in each layer, then uses router logits $R(x)$ to group experts, followed by frequency-based merging. Nevertheless, in task-agnostic settings without retraining, relying on frequency information for clustering proves ineffective in Table 2 and Table 3. This approach faces three main issues. First, frequency varies across tasks, as shown in Appendix E, making it an unreliable indicator for deciding how many experts to retain in each layer. Second, high-frequency experts within the same layer are rarely merged, overlooking their functional similarities in the feature space. Moreover, grouping

based on router information can be problematic, as it depends on dataset-dependent statistics. Together, the limitations can potentially hinder the model's ability to maintain performance over diverse tasks without access to task data.

## 3. Methodology

### 3.1. Problem Definition

In this study, we address the challenge of reducing the space complexity of an SMoE model through a process termed *expert merging*. This process consolidates existing experts in an SMoE layer into a smaller set while preserving the model's performance. Each SMoE layer initially contains $n$ experts, as defined in Section 2.1. We aim to merge these experts into $r$ clusters, where $r$ represents the target number of experts after merging. For the $i$-th cluster, denoted as $C_i = \{E_0^i, E_1^i, \ldots, E_{|C_i|}^i\}$, $|C_i|$ represents the number of original experts assigned to this cluster. Unlike conventional model merging (Yadav et al., 2023; Xu et al., 2024) with predefined element combinations, expert merging in an SMoE necessitates a two-phase procedure due to its flexible solution space: first grouping experts into clusters, then merging within each cluster. During the merging phase, experts within each cluster combine into a single new expert, which reduces the total number of experts to $r$. The distribution of original experts across clusters satisfies $\sum_{i=1}^{r} |C_i| = n$, which ensures that all original experts are accounted for in the merging process. The grouping phase encompasses two distinct strategies. Static grouping maintains exactly $r$ experts per layer after merging, while dynamic grouping permits variable expert numbers per layer with an average of $r$. HC-SMoE implements static grouping, aligning with *O-prune*, whereas *F-prune* and *S-prune* in Table 1 utilize dynamic grouping. The router network $R$ after expert merging remains unchanged throughout this process, with inputs previously routed to any expert within a merged group now directed to the corresponding merged expert, as illustrated in Fig. 3. This preserves router dimensionality while achieving expert reduction. The expert merging problem presents several unique challenges absent in conventional model merging. First, experts exhibit high-dimensional, task-specific functional overlap that cannot be trivially disentangled through parameter-space similarity alone. Second, the lack of predefined grouping criteria necessitates clustering methods for determining which subsets of experts can be grouped in a manner that minimizes performance loss in the model. Since training an MoE model demands substantial GPU memory, we address this problem without retraining and utilize a non-benchmark dataset to collect information for the expert merging process as calibration dataset.

## 3.2. Hierarchical Clustering for Sparsely Activated MoE

In light of the challenges from expert clustering and merging, we introduce a new framework for SMoE model compression through expert parameter merging. Our approach achieves the balance between model size reduction, performance retention, and computational efficiency without retraining requirements. It advances beyond previous approaches by addressing three crucial aspects of SMoE compression: (1) expert similarity metrics, (2) grouping methodology, and (3) merging strategies. We present that utilizing averaged expert outputs as similarity metrics (Section 3.2.1) preserves functional equivalence among grouped experts more effectively than router-space comparisons. We then demonstrate that our hierarchical clustering approach (Section 3.2.2) enables progressive expert grouping with reduced sensitivity to initialization variations. Finally, we elaborate on the expert merging process and demonstrate that effective clustering enables our method to preserve the capabilities of the original model across diverse merging strategies (Section 3.2.3). The integration of these components enables HC-SMoE to produce reliable and stable cluster quality, and superior in addressing compression-performance trade-off.

### 3.2.1. EXPERT OUTPUTS AS SIMILARITY METRIC

The primary objective of expert merging process is to minimize functional divergence between the compressed and original models. M-SMoE falls short in this objective as it adopts router-logits as the similarity metric, which leads to the deviation of the function outputs. Motivated by evidence that output similarity correlates with functional equivalence (Li et al., 2016; Stoica et al., 2024), we propose utilizing average expert outputs over a calibration dataset $\mathcal{D}_{cal}$ with $T$ tokens to address the issue. Specifically, for expert $E_j$, the representative vector computation follows:

$$o_j := \mathbb{E}_{x \sim \mathcal{D}_{cal}}[E_j(x)] = \frac{1}{T} \sum_{x \in \mathcal{D}_{cal}} E(x). \quad (4)$$

This formulation captures both contextual input information and expert learned transformations, validated by L2 errors of last layer outputs presented in Table 23 in the Appendix.

Prior arts face fundamental limitations in this context. For instance, router logits $R(x)$ capture input-dependent assignment preferences rather than intrinsic expert functionality, which creates task-specific biases that hinder generalization. Parameter-space comparisons, such as concatenating flattened weights (e.g., $W_{\text{gate}}, W_{\text{down}}, W_{\text{up}}$), face computational inefficiency and require $O(3d^2)$ operations for $d$-dimensional experts and parameter redundancy in SMoE structures, as shown by Liu et al. (2023). In contrast, our average-output-based metric operates in $O(d)$ space with reduced memory overhead, aligns directly with the merging objective, and maintains consistency across input distribution shifts. This property ensures robustness across diverse

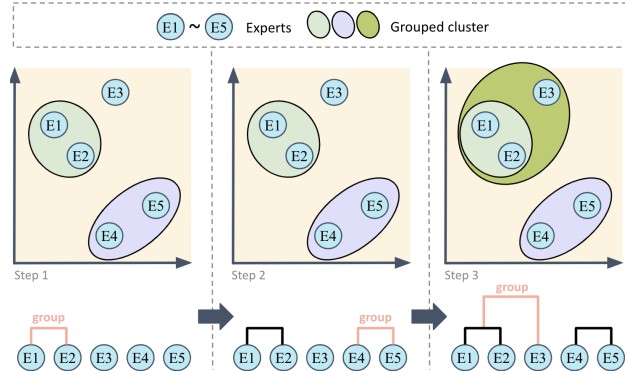

*Figure 2.* Illustration of the proposed hierarchical clustering strategy based on expert outputs. Each blue circle denotes the outputs of an expert in the embedding space. Hierarchical clustering would iteratively group the expert clusters with minimum cluster distance.

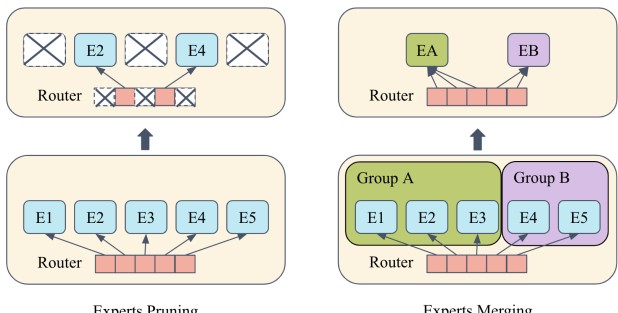

*Figure 3.* Comparison of expert pruning and merging strategies.

tasks, as validated in Section 4.3, where HC-SMoE outperforms router-logits- and weight-based baselines. Note that additional quantitative results are available in Appendix D.

### 3.2.2. HIERARCHICAL CLUSTERING OF EXPERTS

With a reliable expert similarity metric established, the subsequent step involves clustering SMoE experts into $r$ groups for the merging process. To achieve this objective, we employ Hierarchical Clustering (HC) as the core mechanism for grouping experts based on its capability to dynamically adapt cluster assignments while maintaining initialization robustness. Unlike static partitioning methods, HC combines experts through a bottom-up agglomerative process: starting with each expert as a singleton cluster, it recursively combines the most functionally similar pairs while continuously recalculating inter-cluster distances. This iterative recalibration reflects current functional affinities of evolving clusters and enables adaptation to emergent behaviors, a capability absent in static partitioning based approaches.

Prior approaches suffer from fundamental limitations in this context. The one-shot grouping strategy proposed by Li et al. (2024), for instance, freezes cluster assignments after an initial partitioning, disregarding how merged experts alter the functional landscape of remaining clusters. Similarly,

K-means (Ikotun et al., 2023) imposes restrictive assumptions about cluster geometry (e.g., spherical, equally sized) and exhibits sensitivity to centroid initialization, a flaw amplified in high-dimensional expert output spaces where random initialization often traps clusters in suboptimal local minima (Table 5 and Appendix D). HC circumvents these issues through its dendrogram-based structure, which accommodates heterogeneous cluster shapes and sizes without requiring explicit assumptions. HC recursively applies deterministic linkage criteria (e.g., average linkage) to preserve hierarchical expert relationships while mitigating initialization dependencies. The clustering process requires two essential components: (1) a distance metric for measuring differences between expert output vectors, and (2) a linkage strategy for determining inter-cluster distances. Our implementation uses the Euclidean distance, expressed as:

$$d(e_i, e_j) = ||e_i - e_j||_2 \tag{5}$$

where $e_i$ and $e_j$ represent the metric values for computing distances between experts $i$ and $j$. For the linkage strategy, we investigate three methods: *single*, *complete*, and *average*:

$$\text{single:} \quad \min_{a \in A, b \in B} d(a, b), \tag{6}$$

$$\text{complete:} \quad \max_{a \in A, b \in B} d(a, b), \tag{7}$$

$$\text{average:} \quad \frac{1}{|A| \cdot |B|} \sum_{a \in A} \sum_{b \in B} d(a, b), \tag{8}$$

where $A$ and $B$ represent clusters, and $a$ and $b$ denote experts that belong to these clusters. Single linkage defines cluster distances through the closest pair of elements, while complete linkage uses the maximum distance and often produces overly compact clusters that miss subtle similarities. Average linkage considers the mean pairwise distance between cluster elements and achieves an optimal balance. As a result, the proposed HC-SMoE framework employs average-linkage HC to optimize the trade-off between intra-cluster homogeneity and inter-cluster distinctiveness. The theoretical justification is provided in Appendix A.

### 3.2.3. EXPERT MERGING

Following cluster formation, HC-SMoE merges experts within each cluster via a parametrized weight-space aggregation. Our empirical evidence indicates that while the choice of merging method does influence the overall performance, its impact is relatively modest compared to the significance of clustering results. Specifically, HC-SMoE merges the clustered experts on the weight space as: $\hat{E}_i = \sum_{j=1}^{|C_i|} \alpha_j E_j$, where $\sum_{j=1}^{|C_i|} \alpha_j = 1$, and $\alpha_j$ denotes the weight for merging expert $j$. This study considers three different merging strategies: *average* merging, *frequency-weighted* merging, as well as *fixed-dominant* merging. In average merging, $\alpha_j = \frac{1}{|C_i|}$. In frequency-weighted merging, $\alpha_j$ denotes the usage frequency of expert $j$. On the other hand, fixed-dominant merging, a methodology introduced in this study, represents an efficient adaptation of ZipIt specifically developed for merging experts in SMoE models. As developing novel merging methods extends beyond the focus of this article, we present the comprehensive analysis of fixed-dominant merging and comparisons with ZipIt in Appendix B.2. HC-SMoE implements frequency-weighted merging to maintain merging strategy flexibility, while preserving the task-agnostic nature of hierarchical clustering. The experimental results presented in Table 7 further reveal that merging strategy selection has marginal impact when utilizing a general-purpose calibration dataset.

## 4. Experimental Results

### 4.1. Experimental Settings

We conduct experiments on two SMoE models: Qwen1.5-MoE-A2.7B (henceforth Qwen) (Team, 2024) and Mixtral 8x7B (Jiang et al., 2024). For Qwen, we explore two levels of reduction: merging the number of experts from 60 to 45 and further to 30 per layer. This corresponds to a reduction in parameters from 14.3B to 11.2B (denoted as Qwen 45x2.7B), and subsequently to 8.1B (denoted as Qwen 30x2.7B). Similarly, Mixtral 8x7B undergoes reduction from eight to six experts and then to four experts per layer, decreasing the total parameters from 46.7B to 35.6B (denoted as Mixtral 6x7B) and further to 24.3B (denoted as Mixtral 4x7B). This graduated approach enables the evaluation of expert merging impact at different levels of model reduction. Experiments on Mixtral 8x7B and Qwen are conducted on eight NVIDIA A100 GPUs and four NVIDIA V100 GPUs, respectively.

To evaluate our method in a task-agnostic setting, we utilize eight tasks using the EleutherAI Language Model Evaluation Harness (Gao et al., 2024). These are designed to cover various aspects of language understanding and reasoning, including both Challenge and Easy sets in AI2 Reasoning Challenge (ARC) (Clark et al., 2018), BoolQ (Clark et al., 2019), HellaSwag (Zellers et al., 2019), Massive Multi-task Language Understanding (MMLU) (Hendrycks et al., 2021a), OpenBookQA (Mihaylov et al., 2018), Recognizing Textual Entailment (RTE) (Bentivogli et al., 2009) and Winograd Schema Challenge (Sakaguchi et al., 2021). We report zero-shot accuracy on those benchmarks. To demonstrate HC-SMoE's domain adaptability, additional evaluations on medical reasoning tasks (are offered in Appendix B.4.2).

For our comparisons, three pruning baselines are employed: *O-prune* (Lu et al., 2024), *S-prune* (He et al., 2024), and *F-prune*. *F-prune*, where '*F*' denotes frequency, adheres to the same methodology as *S-prune*. However, it employs frequency as the criterion for pruning experts, in contrast to *S-prune* which utilizes router logits. Due to the high

*Table 2.* Zero-shot comparison of Qwen1.5-MoE-A2.7B-Chat: original architecture v.s. reduced versions with 45 and 30 experts per layer. **HC-SMoE (avg)** stands for average linkage when performing hierarchical clustering. **HC-SMoE (single)** stands for single linkage.

| Model | Method | ARC-c | ARC-e | BoolQ | HellaSwag | MMLU | OBQA | RTE | Winogrande | Average |
|---|---|---|---|---|---|---|---|---|---|---|
| Qwen 60x2.7B | None | 0.3951 | 0.7012 | 0.8135 | 0.5932 | 0.6047 | 0.310 | 0.7329 | 0.6559 | 0.6008 |
| Qwen 45x2.7B | O-prune ($10^5$) | 0.3268 | 0.6111 | 0.7566 | 0.5388 | 0.5150 | 0.268 | 0.6498 | 0.6330 | 0.5374 |
| | F-prune | 0.3490 | 0.5989 | 0.7618 | 0.5441 | 0.4560 | **0.282** | **0.7690** | 0.6409 | 0.5502 |
| | S-prune | 0.3464 | 0.6061 | 0.7128 | 0.5228 | 0.4930 | 0.264 | 0.6534 | 0.5935 | 0.5240 |
| | M-SMoE | 0.3473 | 0.6157 | 0.7544 | 0.5157 | 0.4182 | 0.262 | 0.7292 | 0.6377 | 0.5350 |
| | HC-SMoE (avg) | **0.3660** | **0.6578** | **0.7948** | 0.5520 | 0.5332 | 0.272 | 0.7509 | 0.6464 | **0.5716** |
| | HC-SMoE (single) | 0.3592 | **0.6578** | 0.7942 | **0.5578** | **0.5360** | 0.270 | 0.7292 | **0.6472** | 0.5689 |
| Qwen 30x2.7B | O-prune ($10^5$) | 0.2568 | 0.4449 | 0.6496 | 0.4351 | 0.2907 | 0.202 | 0.6065 | 0.5375 | 0.4279 |
| | F-prune | 0.2765 | 0.4718 | 0.6587 | 0.4330 | 0.3023 | 0.230 | 0.6570 | 0.5927 | 0.4528 |
| | S-prune | 0.2500 | 0.4756 | 0.6388 | 0.4041 | 0.3471 | 0.196 | 0.6209 | 0.5146 | 0.4309 |
| | M-SMoE | 0.1945 | 0.2786 | 0.4462 | 0.2837 | 0.2475 | 0.160 | 0.4477 | 0.5185 | 0.3221 |
| | HC-SMoE (avg) | **0.3532** | **0.6149** | 0.7535 | **0.4695** | 0.4534 | **0.228** | **0.6606** | **0.6456** | **0.5223** |
| | HC-SMoE (single) | 0.3524 | 0.6153 | **0.7661** | 0.4661 | **0.4537** | **0.228** | 0.6534 | 0.6306 | 0.5207 |

computational complexity of *O-prune* on Qwen, a random sampling of $10,000$ possible expert sets in each layer is performed instead. The set with the smallest output difference from the original model is selected, denoted as *O-prune* ($10^5$) in the Qwen experiments. In addition, M-SMoE is included as the merging baseline and applied in a task-agnostic setting without retraining to ensure a fair comparison. All baselines and HC-SMoE require a calibration dataset to estimate input statistics. This dataset is constructed by sampling from the C4 corpus (Raffel et al., 2020), concatenating extracted text into 32 sequences of $2,048$ tokens each. To further validate the independence of HC-SMoE from the calibration dataset, we construct two additional datasets from MATH (Hendrycks et al., 2021b) and CodeQA (Liu & Wan, 2021). Please refer to our Appendix B.3 for more details.

### 4.2. Performance Comparisons

This section presents a comprehensive comparison of the performance of the models reduced by HC-SMoE against the original SMoE models and the baselines. The analysis encompasses various model sizes and tasks, and provides insights into the efficacy and scalability of the proposed HC-SmoE method. As presented in Tables 2 and 3, the M-SMoE baseline exhibits the lowest performance across all benchmarks, indicating the ineffectiveness of router-logit-based grouping in a task-agnostic setting. *O-prune* demonstrates suboptimal performance, particularly on Qwen, due to its limitations in evaluating all possible expert combinations. This results in a substantial performance decline compared to Mixtral. In contrast, HC-SMoE demonstrates consistent superiority over these baselines, irrespective of model size, and proves applicable to different numbers of SmoE experts.

Qwen 45x2.7B and Mixtral 4x7B achieve comparable scores despite a twofold difference in parameter count. This obser-

vation substantiates the scalability of HC-SMoE to SMoE models with a higher number of experts. With a 25% reduction in experts, our method even surpasses the original model on certain tasks, such as Mixtral 6x7B on BoolQ and Qwen 45x2.7B on RTE. This improvement can be attributed to the reduction of expert redundancy after merging. In this configuration, both Qwen and Mixtral exhibit an average performance gap of less than 3% compared to their original models. Even with a 50% reduction, HC-SMoE applied to Qwen maintains a gap of merely 7.43% and outperforms the best baseline, *F-prune*, which lags behind HC-SMoE by 7.46%. The experimental results validate the robustness and efficacy of HC-SMoE over diverse model sizes and tasks.

### 4.3. Ablation Study

**Ablation on Different Linkage Methods among Different Metrics.** Table 4 presents a comparison of different linkage methods in hierarchical clustering according to various metrics: *router-logits*, *weight*, and *expert-output*. Hierarchical clustering exhibits stability due to its deterministic nature. This stability is evidenced by consistent performance across benchmarks and the highest average scores. Unlike K-means, it is not susceptible to initialization randomness, which establishes it as a more reliable clustering method. Among the different linkage methods, single linkage generally performs satisfactorily. However, average linkage emerges as the superior option and achieves the highest scores in most of the evaluated settings. The experimental results further reveal an interesting pattern in the performance of complete linkage across different metrics. When applied with the expert-output metric, complete linkage yields suboptimal results, achieving only $0.3909$ on average. The performance further deteriorates with the weight metric, which reaches a mere $0.3682$. On the contrary, the

*Table 3.* Zero-shot comparison of Mixtral 8x7B: original architecture v.s. reduced versions with six and four experts per layer.

| Model | Method | ARC-c | ARC-e | BoolQ | HellaSwag | MMLU | OBQA | RTE | Winogrande | Average |
|---|---|---|---|---|---|---|---|---|---|---|
| Mixtral 8x7B | None | 0.5648 | 0.8422 | 0.8505 | 0.6490 | 0.6712 | 0.350 | 0.7112 | 0.7593 | 0.6748 |
| Mixtral 6x7B | O-prune | **0.5205** | 0.8009 | 0.8352 | 0.6115 | 0.5741 | 0.316 | 0.6606 | **0.7719** | 0.6363 |
| | F-prune | 0.5009 | 0.7904 | 0.7725 | 0.5990 | 0.5099 | 0.326 | 0.5596 | 0.7672 | 0.6032 |
| | S-prune | 0.4991 | 0.7891 | 0.7801 | 0.5984 | 0.5103 | **0.340** | 0.5704 | 0.7735 | 0.6076 |
| | M-SMoE | 0.2619 | 0.5564 | 0.5208 | 0.4320 | 0.2503 | 0.194 | 0.5271 | 0.5848 | 0.4159 |
| | HC-SMoE (avg) | 0.5145 | 0.8043 | **0.8554** | 0.6142 | 0.6043 | 0.324 | **0.6715** | 0.7514 | **0.6425** |
| | HC-SMoE (single) | 0.5154 | **0.8123** | **0.8554** | **0.6163** | **0.6053** | 0.310 | **0.6715** | 0.7403 | 0.6408 |
| Mixtral 4x7B | O-prune | 0.4394 | 0.7327 | 0.8046 | 0.5660 | 0.4584 | 0.286 | 0.5668 | **0.7285** | 0.5728 |
| | F-prune | 0.4352 | 0.7290 | 0.7520 | 0.5293 | 0.3739 | **0.290** | 0.5560 | 0.7245 | 0.5487 |
| | S-prune | 0.2235 | 0.4339 | 0.6300 | 0.4250 | 0.2554 | 0.188 | 0.5235 | 0.5699 | 0.4062 |
| | M-SMoE | 0.2116 | 0.2765 | 0.4954 | 0.2767 | 0.2452 | 0.108 | 0.4910 | 0.4964 | 0.3251 |
| | HC-SMoE (avg) | 0.4573 | 0.7454 | 0.8018 | 0.5709 | 0.4571 | 0.270 | 0.5523 | **0.7285** | 0.5729 |
| | HC-SMoE (single) | **0.4642** | **0.7483** | **0.8321** | **0.5781** | **0.4895** | 0.280 | **0.5884** | 0.7206 | **0.5877** |

*Table 4.* Different linkage method comparisons of hierarchical clustering on Qwen 45x2.7B, where 'rl' denotes router-logits and 'eo' denotes expert-output.

| Linkage | Metric | ARC-c | BoolQ | OBQA | RTE | Average |
|---|---|---|---|---|---|---|
| None | None | 0.3951 | 0.8135 | 0.310 | 0.7329 | 0.5629 |
| Single | rl | 0.2398 | 0.3792 | 0.180 | 0.5054 | 0.3261 |
| | weight | 0.3695 | 0.7676 | 0.254 | 0.7004 | 0.5229 |
| | eo | 0.3592 | 0.7942 | 0.270 | 0.7292 | 0.5382 |
| Complete | rl | 0.3677 | 0.7694 | 0.248 | 0.7329 | 0.5295 |
| | weight | 0.2363 | 0.4446 | 0.178 | 0.6137 | 0.3682 |
| | eo | 0.2338 | 0.6037 | 0.210 | 0.5162 | 0.3909 |
| Average | rl | 0.2073 | 0.3801 | 0.172 | 0.5018 | 0.3153 |
| | weight | **0.3788** | 0.7645 | 0.250 | 0.7004 | 0.5234 |
| | eo | 0.3660 | **0.7948** | **0.272** | **0.7509** | **0.5459** |

*Table 5.* Performance comparison between the proposed HC-SMoE and K-means clustering approaches on Qwen 30x2.7B. K-means-**fix** designates the first $r$ experts as initial centers, while K-means-**rnd** randomly selects $r$ experts as initial centers. Note that the best performance across all methods are marked in **bold**, with the best performance in each clustering method marked with underline.

| Cluter | Metric | ARC-c | BoolQ | OBQA | RTE | Average |
|---|---|---|---|---|---|---|
| K-fix | rl | 0.2031 | 0.4015 | 0.162 | 0.4838 | 0.3126 |
| | weight | 0.2073 | 0.4960 | 0.166 | 0.509 | 0.3446 |
| | eo | 0.2184 | 0.3786 | 0.148 | 0.5343 | 0.3198 |
| K-rnd | rl | 0.2014 | 0.4168 | 0.142 | 0.5018 | 0.3155 |
| | weight | 0.2108 | 0.533 | 0.174 | 0.5379 | 0.3639 |
| | eo | 0.3370 | 0.6398 | 0.224 | 0.6065 | 0.4518 |
| HC | eo | **0.3532** | **0.7535** | **0.228** | **0.6606** | **0.4988** |

router-logits-based approach excels exclusively with complete linkage, and attains an average score of $0.5295$. This disparity substantiates the distinctive properties of router-logits compared to weights and expert outputs. This observation can be attributed to the inherent characteristics of the similarity metrics. Router-logits align well with complete linkage since they capture the maximal boundaries between clusters. This alignment effectively reflects distinct activation patterns. In contrast, expert outputs and weights benefit from single or average linkage methods. These metrics reveal more subtle, internal similarities that may not manifest through extreme distances. Therefore, they favor linkage methods that consider average or minimal distances between cluster elements.

**K-means Clustering v.s. Hierarchical Clustering**. We next present a comparative analysis between our hierarchical clustering (HC) method and various K-means clustering strategies, underscoring the superiority of HC. Table 5 reports the performance of different initialization strategies and similarity metrics in K-means, evaluated across four benchmarks: ARC-c, BoolQ, OBQA, and RTE. These benchmarks were selected for their comprehensive cover-

age of language abilities, encompassing common sense reasoning, basic knowledge questions, and semantic similarity between sentence pairs. The evaluation results reveal that most post-merged models utilizing K-means experience a substantial decline in their original capabilities. For instance, even the best-performing model employing the expert-output similarity metric achieves a score 4.75% lower than our HC-SMoE results. This performance gap validates the effectiveness of our proposed HC-based method.

K-means also exhibits significant instability, particularly when juxtaposed with HC. The final performance of K-means demonstrates high sensitivity to the choice of initial cluster centers. In experiments conducted on the Qwen45x2.7B model using the weight similarity metric, we observe a substantial average accuracy reduction of 12.96% when transitioning from a fixed to a random initialization strategy. This sensitivity illuminates K-means' inherent randomness and lack of robustness. The observed instability and performance degradation in K-means clustering fur-

*Table 6.* Comparisons for different similarity metric to single-shot grouping method and our HC-SMoE on Mixtral 8x7B when reducing SMoE experts to average six and four per layer.

| Model | Metric | ARC-c | ARC-e | BoolQ | HellaSwag | MMLU | OBQA | RTE | Winogrande | Average |
|---|---|---|---|---|---|---|---|---|---|---|
| Mixtral 8x7B | None | 0.5648 | 0.8422 | 0.8505 | 0.6490 | 0.6712 | 0.350 | 0.7112 | 0.7593 | 0.6748 |
| Mixtral 6x7B | *router-logits* | 0.2619 | 0.5564 | 0.5208 | 0.432 | 0.2503 | 0.194 | 0.5271 | 0.5848 | 0.4159 |
| | *weight* | 0.4974 | 0.7955 | 0.7810 | 0.6131 | 0.5244 | 0.340 | **0.6715** | **0.7585** | 0.6227 |
| | *expert-output* | 0.5060 | **0.8056** | 0.8373 | 0.6130 | 0.5595 | 0.306 | 0.6318 | 0.7474 | 0.6258 |
| | HC-SMoE | **0.5145** | 0.8043 | **0.8554** | **0.6142** | **0.6043** | 0.324 | **0.6715** | 0.7514 | **0.6425** |
| Mixtral 4x7B | *router-logits* | 0.2116 | 0.2765 | 0.4954 | 0.2767 | 0.2452 | 0.108 | 0.4910 | 0.4964 | 0.3251 |
| | *weight* | 0.4172 | 0.7382 | 0.7862 | 0.5457 | 0.4223 | 0.256 | 0.5523 | 0.7143 | 0.5540 |
| | *expert-output* | 0.4326 | 0.7386 | **0.8021** | 0.5467 | 0.4290 | 0.278 | 0.5704 | 0.7245 | 0.5652 |
| | HC-SMoE | **0.4573** | **0.7454** | 0.8018 | **0.5709** | **0.4571** | 0.270 | 0.5523 | **0.7285** | **0.5729** |

*Table 7.* Various merging methods with HC average linkage based on expert outputs. **Fix-Dom** represents fixed-dominant merging described in Section 3.2.3. **Avg** in the Merge column denotes the average score among all the merging strategy under same model settings.

| Model | Merge | ARC-c | ARC-e | BoolQ | HellaSwag | MMLU | OBQA | RTE | Winogrande | Average |
|---|---|---|---|---|---|---|---|---|---|---|
| Qwen 60x2.7B | None | 0.3951 | 0.7012 | 0.8135 | 0.5932 | 0.6047 | 0.310 | 0.7329 | 0.6559 | 0.6008 |
| Qwen 45x2.7B | Frequency | 0.3660 | 0.6578 | 0.7948 | 0.5520 | 0.5332 | **0.272** | **0.7509** | 0.6464 | **0.5716** |
| | Average | 0.3584 | 0.6553 | **0.7936** | 0.5516 | **0.5348** | 0.270 | 0.7473 | **0.6559** | 0.5709 |
| | Fix-Dom | **0.3695** | **0.6692** | 0.7896 | **0.5555** | 0.5338 | 0.262 | 0.7365 | 0.6535 | 0.5712 |
| | Avg | 0.3646 | 0.6608 | 0.7927 | 0.5530 | 0.5339 | 0.268 | 0.7449 | 0.6519 | 0.5712 |
| Qwen 30x2.7B | Frequency | 0.3532 | **0.6149** | 0.7535 | **0.4695** | **0.4534** | 0.228 | 0.6606 | 0.6456 | 0.5223 |
| | Average | **0.3575** | 0.6145 | **0.7554** | 0.4706 | 0.4531 | 0.228 | **0.6643** | 0.6488 | **0.5240** |
| | Fix-Dom | 0.3439 | 0.6132 | 0.7544 | 0.4679 | 0.4445 | 0.228 | **0.6643** | **0.6504** | 0.5208 |
| | Avg | 0.3515 | 0.6142 | 0.7544 | 0.4693 | 0.4503 | 0.228 | 0.6631 | 0.6483 | 0.5224 |

ther accentuate the stability and efficacy of our HC-based method. These findings reinforce the superiority of HC in maintaining model performance post-merging and its resilience to initialization variability.

**Single-shot Grouping v.s. Hierarchical Clustering**. In this analysis, we follow the single-shot grouping methods outlined in (Li et al., 2024) to compare results on Mixtral 8x7B, and report the results in Table 6. Among the similarity metrics evaluated, *router-logits* exhibits the poorest performance, indicating its unsuitability for task-agnostic settings due to its reliance on dataset-specific statistics. In both the 25% and 50% parameter reduction scenarios, all one-shot grouping methods underperform compared to *O-prune* presented in Table 3. This observation suggests that these grouping methods fail to form effective clusters, and can potentially result in lower performance even when attempting to absorb all expert knowledge. The method based on the expert output metric demonstrates superior performance over other similarity metrics. It outperforms router-logits by 24.01% and weights by 1.12% when reducing 50% of the expert parameters. This finding highlights the importance of selecting appropriate similarity metrics for effective expert grouping. The results reveal that HC-SMoE demonstrates a clear advantage over the one-shot grouping approaches. It achieves average improvements of 1.98% and 1.67% in the 25% and 50% parameter reduction settings, respectively.

**Ablation on Different Merging Methods.** Table 7 presents the results of hierarchical clustering with three merging strategies: *frequency*, *average*, and *fixed-dominant merging*. For Qwen30x2.7B, the average merging method demonstrates superior performance. It exceeds frequency merging by 0.17% and marginally enhances overall performance. This outcome substantiates our assertion that once a high-quality cluster is identified, the specific merging method becomes modestly influential on the final performance. The rationale behind this phenomenon lies in the functional similarity exhibited by experts within the same group, as evidenced by their similar outputs. Thus, the model maintains robust performance irrespective of the merging strategy employed. It is noteworthy that all three merging methods outperform the four baselines in Table 2. This observation further substantiates the effectiveness of HC-SMoE in preserving model performance during the merging process.

# 5. Conclusion

In this paper, we presented HC-SMoE, a retraining-free, task-agnostic, and scalable expert merging framework that employed hierarchical clustering to reduce the parameters of SMoE models. By employing on expert outputs as the similarity metric and leveraging hierarchical clustering, HC-SMoE effectively captured functional similarities between experts, surpassing previous merging and pruning methods. Our comprehensive evaluation on two representative large-scale models, Qwen and Mixtral, demonstrated that HC-SMoE retained the models' general language abilities even when significantly reducing the number of experts. The experimental results also validated the robustness and scalability of our approach. HC-SMoE achieved notable improvements over existing baselines. This work not only provided a practical solution for optimizing SMoE models but also opened up a broader domain for further research on task-agnostic model compression strategies for SMoE.

## Impact Statement

In compliance with ICML's guidelines, we affirm that our HC-SMoE algorithm does not involve human subjects, personal data, or any components that could raise ethical concerns. The algorithm focuses solely on clustering machine learning model experts based on output similarity to optimize parameter efficiency, ensuring no ethical issues are present.

## Acknowledgements

The authors gratefully acknowledge support from the National Science and Technology Council (NSTC) in Taiwan under grant numbers MOST 111-2223-E-002-011-MY3, NSTC 113-2221-E-002-212-MY3, and NSTC 113-2640-E-002-003. We also express our sincere appreciation to NVIDIA Corporation and the NVIDIA AI Technology Center (NVAITC) for the donation of GPUs and access to the Taipei-1 supercomputer. Furthermore, we thank the National Center for High-Performance Computing for providing computational and storage resources.

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

## Appendix: Retraining-Free Merging of Sparse MoE via Hierarchical Clustering

## A. Theoretical Justification

In this section, we present a theoretical rationale for the HC-SMoE algorithm, which employs hierarchical clustering with unweighted average linkage to group experts into clusters and reduce the number of parameters in a Mixture-of-Experts (MoE) model. Our objective is to minimize the approximation error between the original MoE output and the HC-SMoE output while leveraging the performance guarantees for hierarchical clustering.

Let $\{G_j\}_{j=1}^r$ denote a partition sets of the expert indices into $r$ clusters. We define the average-merged expert for group $j$ as:

$$\bar{E}_j(x) := \frac{1}{|G_j|} \sum_{i \in G_j} E_i(x), \tag{9}$$

where $i$ represents the expert index and $g(i)$ is the function that maps expert $E_i$ to its cluster index $j = g(i)$. The HC-SMoE output can be expressed as:

$$y_{\text{HC}}(x) = \sum_{j=1}^r (\sum_{i \in G_j} P_i(x)) \cdot \bar{E}_j(x) = \sum_{i=1}^n P_i(x) \cdot \bar{E}_{g(i)}(x), \tag{10}$$

where $P_i(x)$ denotes the routing probability assigned to expert $E_i$ for input $x$, and $n$ represents the total number of experts. The approximation error between the original MoE output and the HC-SMoE output is formulated as:

$$\|y_{\text{orig}}(x) - y_{\text{HC}}(x)\|^2 = \|\sum_{i=1}^n P_i(x)(E_i(x) - \bar{E}_{g(i)}(x))\|^2 \leq \sum_{i=1}^n P_i(x) \cdot \|E_i(x) - \bar{E}_{g(i)}(x)\|^2, \tag{11}$$

by Jensen's inequality applied to the convex function $\|\cdot\|^2$. This inequality demonstrates that the HC-SMoE approximation error is bounded by the weighted intra-cluster variance. To reduce the approximation error $\|y_{\text{orig}}(x) - y_{\text{HC}}(x)\|^2$, we opt to to minimize the intra-cluster variance $\|E_i(x) - \bar{E}_{g(i)}(x)\|^2$.

### A.1. Clustering Optimality and Approximation Bound

Finding the optimal solution, defined as $\text{OPT} := \min \|E - \bar{E}\|^2$, is an NP-hard problem. A naive approach would involve exhaustively evaluating all possible partitions of the $n$ experts into $r$ clusters, where the total number of such partitions is given by:

$$\frac{1}{r!} \sum_{i=0}^r (-1)^i \binom{r}{i} (r-i)^n. \tag{12}$$

Hierarchical clustering with average linkage is a polynomial-time algorithm that provides an approximation to the optimal clustering. In the worst case, it guarantees a solution satisfying $\|E - \bar{E}\|^2 \leq 3 \cdot \text{OPT}$ (Moseley & Wang, 2023). Therefore, we adopt this computationally efficient method in place of the naive exhaustive approach.

Furthermore, unlike K-means and K-center which depend on random initialization, HC operates as a deterministic algorithm. This deterministic nature ensures stable results across executions. Our empirical evidence indicates that HC with average linkage performs effectively in practice, and is able to deliver robust clustering results without randomness requirements, which benefits parameter reduction in MoE models.

### A.2. HC-SMoE Algorithm

The HC-SMoE algorithm performs hierarchical clustering on expert outputs and merges experts within each cluster based on frequency-weighted averaging.

---

**Algorithm 1** HC-SMoE: Hierarchical Clustering for Sparse Mixture-of-Experts

---

**Require:** Calibration dataset $\mathcal{D}_{cal}$ with $T$ tokens, number of experts within an MoE layer $n$, set of experts $\mathbb{E} = \{E_1, \ldots, E_n\}$, target number of clusters $r$.

**Ensure:** Merged expert set $\{\bar{E}_1, ..., \bar{E}_r\}$.

1: // Collect averaged expert outputs
2: **for** each expert $E_i \in \mathbb{E}$ **do**
3:     $o_i = \mathbb{E}_{x \sim \mathcal{D}_{cal}}[E_i(x)]$ // Compute mean expert output
4: **end for**
5: // Hierarchical clustering
6: Initialize clusters $\mathcal{C} = \{\{E_1\}, \{E_2\}, ..., \{E_n\}\}$ and set $m \leftarrow n$
7: **while** $m > r$ **do**
8:     Compute pairwise Euclidean distance between expert outputs:

$$d(C_a, C_b) = \frac{1}{|C_a||C_b|} \sum_{o_i \in C_a, o_j \in C_b} \|o_i - o_j\|_2$$

9:     Merge two clusters $(C_a, C_b)$ with smallest $d(C_a, C_b)$ using average linkage.
10:     Update $\mathcal{C} \leftarrow \mathcal{C} \setminus \{C_a, C_b\} \cup \{C_a \cup C_b\}$ and $m \leftarrow m - 1$.
11: **end while**
12: // Frequency-weighted merging
13: **for** each cluster $C_j \in \mathcal{C}$ **do**
14:     Let $f_i$ be the frequency of tokens assigned to expert $E_i$.
15:     Normalize: $\tilde{f}_i = \frac{f_i}{\sum_{E_i \in C_j} f_i}$.
16:     Compute merged expert:

$$\bar{E}_j = \sum_{E_i \in C_j} \tilde{f}_i E_i$$

17: **end for**

---

# B. Exploratory Experiments

## B.1. Non-Uniform Hierarchical Clustering

In our main experiments, the number of clusters in each layer is fixed and uniform due to model design choices. Here, we explore a more flexible approach that allows different numbers of clusters in each layer while maintaining an overall 25% or 50% reduction of experts. To determine the cluster count per layer, we first select the top $r\%$ most frequently activated experts based on their activation frequencies across layers. We then count the number of these experts that remain in each layer to guide the selection of clusters in that layer, followed by hierarchical clustering.

For example, in the uniform clustering setting for Qwen with a 25% reduction, the distribution will be $[45, 45, 45, 45, ..., 45]$ across all layers. In contrast, the non-uniform setting might result in a distribution like $[48, 45, 40, 42, 50, ...]$, as long as the overall number of clusters aligns with the target reduction. Table 8 presents the results of this non-uniform clustering strategy.

## B.2. Fixed-Dominant Merging

The fixed-dominant (Fix-Dom) merging approach modifies the traditional ZipIt (Stoica et al., 2024) feature similarity calculation. Rather than concatenating features from all experts and computing pairwise correlations, we fix the feature order of a designated dominant expert as a reference point. Correlations are then computed between this fixed order and the features of other experts, as shown in Figure 4. Features from secondary experts are grouped with their most correlated counterparts in the dominant expert. The merging process then applies an appropriate weighting scheme, such as average merging, preserving the dominant expert's weight feature order while simplifying the merging process.

Feature similarity is defined as the pairwise correlation between these output features, using formulas adapted from (Li et al., 2016). In the original ZipIt model merging, output features are taken after each linear layer. However, since we aim to merge entire experts, each containing three linear layers, we use the intermediate activation features, which is the activations after the non-linear function and before feeding into $W_{down}$: $\text{act} = (xW_{gate}) \odot xW_{up}$ to compute similarity. This approach considers expert similarity from an activation perspective, but we can also use the experts' weights as the "feature" for correlation or even combine both activation and weight features.

The Fix-Dom merging technique has two main advantages: it preserves the structural integrity of the dominant expert's feature arrangement and accelerates the merging process compared to the original ZipIt method. Instead of iteratively selecting and merging highly correlated features until the target dimension is reached, fix-dom merge performs a more efficient grouping. For example, in Mixtral8x4B, ZipIt takes approximately 725 minutes, while Fix-Dom merge completes in just 7 minutes, making it over 100 times faster. For performance comparisons between ZipIt and fix-dom merge using various feature selections (activation, weight, and activation + weight), refer to Table 9.

## B.3. Calibration Dataset

We evaluated our approach using three different calibration datasets: C4 (Raffel et al., 2020), MATH (Hendrycks et al., 2021b), and CodeQA (Liu & Wan, 2021). These datasets vary in their domain focus, where C4 contains general-purpose

*Table 8.* Zero-shot performance evaluation of non-uniform hierarchical clustering for reducing 25% experts of Qwen. We present the clustering results under single and average linkage with weight and expert-output as similarity metric.

| Linkage | Metric | Merge | ARC-c | ARC-e | BoolQ | HellaSwag | MMLU | OBQA | RTE | Winogrande | Average |
|---------|--------|-------|-------|-------|-------|-----------|------|------|-----|------------|---------|
| Single | *weight* | Freq | 0.2108 | 0.3493 | 0.5086 | 0.4536 | 0.2296 | 0.170 | 0.5596 | 0.5801 | 0.3827 |
| | | Fix-Dom | 0.2133 | 0.3531 | 0.4847 | 0.4588 | 0.2303 | 0.168 | 0.6101 | 0.5714 | 0.3862 |
| | *expert-output* | Freq | 0.3686 | 0.6604 | 0.7960 | 0.5587 | 0.5290 | 0.254 | 0.7401 | 0.6543 | 0.5701 |
| | | Fix-Dom | 0.3660 | 0.6612 | 0.7917 | 0.5564 | 0.5302 | 0.262 | 0.7292 | 0.6527 | 0.5687 |
| Average | *weight* | Freq | 0.2125 | 0.3535 | 0.5024 | 0.4543 | 0.2287 | 0.174 | 0.5560 | 0.5785 | 0.3825 |
| | | Fix-Dom | 0.2116 | 0.3497 | 0.4951 | 0.4565 | 0.2327 | 0.164 | 0.5921 | 0.5738 | 0.3844 |
| | *expert-output* | Freq | 0.3575 | 0.6561 | 0.7933 | 0.5538 | 0.5319 | 0.272 | 0.7365 | 0.6551 | 0.5695 |
| | | Fix-Dom | 0.3558 | 0.6582 | 0.7917 | 0.5558 | 0.5306 | 0.270 | 0.7256 | 0.6559 | 0.5680 |

*Table 9.* The comparison between ZipIt and Fix-Dom merging for reducing Mixtral 8x7B to Mixtral 4x7B under the same expert clustering groups.

| Feature | Merge | ARC-c | ARC-e | BoolQ | HellaSwag | MMLU | OBQA | RTE | Winogrande | Average |
|---|---|---|---|---|---|---|---|---|---|---|
| act | zipit | 0.3959 | **0.6978** | 0.7352 | 0.5350 | 0.4256 | 0.252 | 0.5776 | 0.7080 | 0.5409 |
| | Fix-Dom | **0.4036** | 0.6873 | **0.7951** | **0.5351** | **0.4471** | **0.278** | **0.6462** | **0.7174** | **0.5637** |
| weight | zipit | 0.3959 | 0.7062 | 0.7976 | 0.5376 | 0.4318 | 0.266 | **0.5848** | 0.6993 | 0.5524 |
| | Fix-Dom | **0.4334** | **0.7290** | **0.8009** | **0.5608** | **0.4913** | **0.280** | 0.5596 | **0.7253** | **0.5725** |
| act+weight | zipit | 0.4078 | 0.7146 | **0.8125** | 0.5389 | 0.4364 | **0.270** | **0.5921** | 0.7009 | 0.5592 |
| | Fix-Dom | **0.4283** | **0.7184** | 0.7774 | **0.5501** | **0.4737** | 0.264 | **0.5921** | **0.7388** | **0.5679** |

*Table 10.* Ablation study on choice of calibration dataset of eight zero-shot language tasks on Qwen model.

| Model | Calib-Dataset | ARC-c | ARC-e | BoolQ | HellaSwag | MMLU | OBQA | RTE | Winogrande | Average |
|---|---|---|---|---|---|---|---|---|---|---|
| Qwen 60x2.7B | - | 0.3933 | 0.6982 | 0.8119 | 0.5938 | 0.6034 | 0.314 | 0.7401 | 0.6606 | 0.6029 |
| Qwen 45x2.7B | C4 | 0.3660 | 0.6578 | 0.7948 | 0.5520 | **0.5332** | 0.272 | 0.7509 | **0.6464** | 0.5716 |
| | MATH | **0.3746** | **0.6662** | 0.7917 | **0.5478** | 0.5293 | **0.284** | 0.7437 | 0.6425 | **0.5725** |
| | CodeQA | 0.3490 | 0.6578 | **0.7976** | 0.5275 | 0.5194 | 0.260 | **0.7581** | 0.6448 | 0.5643 |
| Qwen 30x2.7B | C4 | **0.3532** | **0.6149** | 0.7535 | **0.4695** | **0.4534** | 0.228 | 0.6606 | **0.6456** | **0.5223** |
| | MATH | 0.2978 | 0.5513 | 0.7593 | 0.4151 | 0.4415 | **0.244** | **0.6823** | 0.6164 | 0.5010 |
| | CodeQA | 0.3089 | 0.5720 | **0.7633** | 0.4179 | 0.4412 | 0.214 | 0.6606 | 0.6069 | 0.4981 |

tasks and closely aligned with language tasks, MATH focused on math question answering, and CodeQA addresses Python code question answering.

The results presented in Table 10 and Table 11 demonstrate that the performance across the eight evaluated language tasks remains consistent, regardless of the calibration dataset used. Although C4 aligns most closely with general language tasks, the clustering results remain stable even when using domain-specific datasets such as MATH or CodeQA. This finding indicates that for general language tasks, the choice of calibration dataset has only a negligible impact on the effectiveness of our method.

## B.4. HC-SMoE on Various Models and Domains

### B.4.1. DEEPSEEK-MOE

We evaluate HC-SMoE on the DeepSeek-MoE-16B-Base model[1] with pruning ratios of 12.5%, 25%, 37.5%, and 50%. In addition to the first layer, the model originally contains 64 experts and one shared expert per layer. When calculating similarity and merging experts, we only consider those 64 experts in the layer. Reducing the number of experts in MoE models introduces complexity due to interactions among neighboring experts within the same layer and across adjacent layers. Our results demonstrate that even after removing 50% of the experts, the model maintains its performance on language benchmarks and retains much of its knowledge, including a well-preserved RTE score. With our method, the expert merging process is completed within ten minutes on a single NVIDIA A100 GPU.

### B.4.2. MEDMCQA

To evaluate HC-SMoE on more complex domain-specific tasks, we conducted additional experiments on MedMCQA (Pal et al., 2022), a challenging medical question-answering dataset. Table 15 presents the experimental results, which demonstrate HC-SMoE's robust performance in domain-specific applications. The performance of HC-SMoE exceeds all three baseline methods in this specialized domain.

---

[1] https://huggingface.co/deepseek-ai/deepseek-moe-16b-base

*Table 11.* Ablation study on choice of calibration dataset of eight zero-shot language tasks on Mixtral model.

| Model | Calib-Dataset | ARC-c | ARC-e | BoolQ | HellaSwag | MMLU | OBQA | RTE | Winogrande | Average |
|-------|---------------|-------|-------|-------|-----------|------|------|-----|------------|---------|
| Mixtral 8x7B | - | 0.5648 | 0.8422 | 0.8505 | 0.6490 | 0.6712 | 0.350 | 0.7112 | 0.7593 | 0.6748 |
| Mixtral 6x7B | C4 | 0.5145 | **0.8043** | **0.8554** | 0.6142 | 0.6043 | **0.324** | 0.6715 | 0.7514 | 0.6425 |
| | MATH | 0.5102 | 0.7992 | 0.8547 | **0.6178** | 0.6026 | 0.322 | 0.6643 | **0.7561** | 0.6409 |
| | CodeQA | **0.5196** | 0.7896 | 0.8456 | 0.6104 | **0.6152** | 0.316 | **0.7256** | **0.7561** | **0.6473** |
| Mixtral 4x7B | C4 | **0.4573** | 0.7454 | 0.8018 | **0.5709** | 0.4571 | 0.270 | 0.5523 | 0.7285 | 0.5729 |
| | MATH | 0.4522 | **0.7546** | 0.8333 | 0.5674 | **0.4985** | **0.292** | 0.5884 | 0.7024 | 0.5861 |
| | CodeQA | 0.4411 | 0.7348 | **0.8416** | 0.5666 | 0.4983 | 0.288 | **0.6390** | **0.7372** | **0.5933** |

*Table 12.* Zero-shot performance evaluation of HC-SMoE (avg) on DeepSeek-MoE-16B-Base with reducing experts to 56, 48, 40, and 32 experts per layer.

| Expert Prune Ratio | ARC-c | ARC-e | BoolQ | HellaSwag | MMLU | OBQA | RTE | Winogrande | Average |
|--------------------|-------|-------|-------|-----------|------|------|-----|------------|---------|
| 0% | 0.4445 | 0.7605 | 0.7251 | 0.5805 | 0.3830 | 0.316 | 0.6390 | 0.7040 | 0.5691 |
| 12.5% | 0.4403 | 0.7588 | 0.7358 | 0.5661 | 0.3531 | 0.206 | 0.6390 | 0.7103 | 0.5512 |
| 25% | 0.4036 | 0.7197 | 0.7339 | 0.5379 | 0.3090 | 0.270 | 0.6390 | 0.6977 | 0.5389 |
| 37.5% | 0.3831 | 0.7033 | 0.7232 | 0.5008 | 0.2882 | 0.248 | 0.6534 | 0.6890 | 0.5236 |
| 50% | 0.3251 | 0.6275 | 0.6810 | 0.4439 | 0.2470 | 0.212 | 0.6426 | 0.6448 | 0.4780 |

MedMCQA comprises a large-scale Multiple-Choice Question Answering (MCQA) dataset designed for real-world medical entrance examinations. The dataset contains over 194,000 high-quality multiple-choice questions from AIIMS and NEET PG entrance exams, which span 2,400 healthcare topics across 21 medical subjects. Our experiments utilize a two-shot prompt format in Table 14. The evaluation protocol assesses whether the model outputs {A, B, C, D} in the subsequent three tokens.

The experimental methodology employs the MedMCQA validation set for evaluation and its training set for calibration. Due to the imbalanced answer distribution in the validation dataset (answer A, B, C, and D, has 950, 735, 617, and 514 samples, respectively), we present a comprehensive analysis through precision, recall, and F1-score metrics.

### B.5. Soft Clustering

To explore soft clustering against hard clustering, we implemented the Fuzzy C-Means (FCM) algorithm (Bezdek et al., 1984), which allows each expert to belong to multiple clusters with varying degrees of membership.

Let $E = e_1, \ldots, e_n$ denote the $n$ experts to be clustered. The FCM algorithm outputs $c$ cluster centers $C = c_1, \ldots, c_c$ and a partition matrix. The degree of membership of expert $e_i$ in cluster $c_j$ is denoted by $u_{ij} \in [0, 1]$. FCM minimizes the following objective function:

$$J_m = \sum_{i=1}^{N} \sum_{j=1}^{C} u_{ij}^m |e_i - c_j|^2, \tag{13}$$

where the membership degrees and cluster centers are updated iteratively as follows:

$$u_{ij} = \left( \sum_{k=1}^{C} \left( \frac{|e_i - c_j|}{|e_i - c_k|} \right)^{\frac{2}{m-1}} \right)^{-1}, \quad c_j = \frac{\sum_{i=1}^{N} u_{ij}^m \cdot e_i}{\sum_{i=1}^{N} u_{ij}^m} \tag{14}$$

For our experiments, we set the hyperparameter $m = 2$.

Nevertheless, applying soft clustering introduces ambiguity in our frequency-weighted merging method. To address this, we modified the merging process. For each cluster, the final merged expert is computed as a weighted sum of the experts, where

*Table 13.* Zero-shot performance evaluation of HC-SMoE (avg) on Mixtral8x7B-Instruct with reducing experts to six and four experts per layer.

| Expert Prune Ratio | ARC-c | ARC-e | BoolQ | HellaSwag | MMLU | OBQA | RTE | Winogrande | Average |
|---|---|---|---|---|---|---|---|---|---|
| 0% | 0.6237 | 0.8708 | 0.8850 | 0.6751 | 0.6822 | 0.364 | 0.7112 | 0.7656 | 0.6972 |
| 25% | 0.5913 | 0.8443 | 0.8749 | 0.6457 | 0.6284 | 0.360 | 0.7329 | 0.7672 | 0.6806 |
| 50% | 0.5162 | 0.7925 | 0.8606 | 0.5995 | 0.5358 | 0.308 | 0.6606 | 0.7395 | 0.6266 |

*Table 14.* Two-shot prompt template of MedMCQA dataset. `<data.question>` will be replaced with the specific question as well as `<data.opa>`, `<data.opb>`, `<data.opc>`, `<data.opd>` will be replaced with options of corresponding question during evluation.

---

Please choose one option among A,B,C,D to answer the question.
Question: Chronic urethral obstruction due to benign prismatic hyperplasia can lead to the following change in kidney parenchyma
Options: A. Hyperplasia B. Hyperophy C. Atrophy D. Dyplasia Ans:C
Question: All of the following are surgical options for morbid obesity except - Options:
A. Adjustable gastric banding B. Biliopancreatic diversion C. Duodenal Switch D. Roux en Y Duodenal By pass Ans:D
Question: `<data.question>` Options: A. `<data.opa>` B. `<data.opb>` C. `<data.opc>` D. `<data.opd>` Ans:

---

the membership degree serves as the weight:

$$e^{c_j} = \sum_{i=1}^{n} u_{ij} e_i \tag{15}$$

Since HC-SMoE does not modify the router weights (as presented in Fig. 3), input tokens are routed to the corresponding merged expert based on their original assignment. In the FCM setting, this direct routing becomes infeasible, as every expert belongs to all clusters to some degree. To adapt, we merged the router weights using the same weighted formula as above.

Table 16 and Table 17 compare HC-SMoE with FCM. The results indicate a significant accuracy degradation when using FCM, which is likely due to interference in the router weights. This performance decline can be attributed to the fundamental differences between hard and soft clustering. In hard clustering, each expert belongs to exactly one cluster, which allows for a clear and unambiguous assignment of input tokens to merged experts. In contrast, soft clustering assigns each expert to multiple clusters with varying degrees of membership, which could lead to a more complex and potentially less effective routing process. Moreover, the weighted merging of router weights in the FCM setting may introduce noise and dilute the specialized knowledge captured by individual experts. This dilution can hinder the model's ability to effectively route input tokens to the most relevant experts, resulting in suboptimal performance.

These findings highlight that applying soft clustering methods in the MoE merging framework requires a more sophisticated design to handle router weights, particularly in retraining-free settings. We consider it an interesting direction for future exploration to develop novel techniques that can leverage the benefits of soft clustering while mitigating the challenges associated with routing and merging in the context of MoE models.

## B.6. Extreme Reduction

To evaluate extreme pruning scenarios, we conducted additional experiments at substantial compression rates of 62.5% and 75%. Our analysis compares HC-SMoE against four established baselines: F-prune, S-prune, O-prune and M-SMoE. The results for the Qwen1.5-MoE-A2.7B-Chat and Mixtral 8x7B models are presented in Table 18 and Table 19, respectively.

Several benchmark tasks employ multiple-choice formats. ARC-c, ARC-e, HellaSwag, MMLU, and OBQA require selection from four options, with 0.25 as the random-guess baseline. BoolQ, RTE, and Winogrande utilize binary choices, with 0.5 as the random-guess baseline. Scores in proximity to these baselines indicate a substantial deterioration of model capabilities.

Under extreme reduction settings, our experiments reveal that all baselines experience significant accuracy degradation, with

*Table 15.* Experimental results of HC-SMoE and three baselines on MedMCQA with Mixtral 8x7B and the compressed version of reducing experts to six and four per layer.

| Model | Method | Accuracy | Precision | Recall | F1 |
|---|---|---|---|---|---|
| Mixtral 8x7B | None | 0.5930 | 0.5876 | 0.5918 | 0.5888 |
| Mixtral 6x7B | F-prune | 0.3615 | 0.4109 | 0.3786 | 0.3498 |
| | S-prune | 0.4794 | 0.4755 | 0.4721 | 0.4703 |
| | M-SMoE | 0.1818 | 0.0909 | 0.1990 | 0.0634 |
| | HC-SMoE (ours) | **0.5018** | **0.4950** | **0.4785** | **0.4828** |
| Mixtral 4x7B | F-prune | 0.3249 | 0.4363 | 0.3197 | 0.2404 |
| | S-prune | 0.0000 | 0.0000 | 0.0000 | 0.0000 |
| | M-SMoE | 0.0160 | 0.0720 | 0.0165 | 0.0263 |
| | HC-SMoE (ours) | **0.3817** | **0.4015** | **0.3883** | **0.3705** |

*Table 16.* Comparison of HC-SMoE and Fuzzy-Cmeans on Qwen1.5-MoE-A2.7B-Chat.

| Model | Method | ARC-c | ARC-e | BoolQ | HellaSwag | MMLU | OBQA | RTE | Winogrande | Average |
|---|---|---|---|---|---|---|---|---|---|---|
| Qwen 60x2.7B | None | 0.3951 | 0.7012 | 0.8135 | 0.5932 | 0.6047 | 0.310 | 0.7329 | 0.6559 | 0.6008 |
| Qwen 45x2.7B | HC-SMoE | **0.3660** | **0.6578** | **0.7948** | **0.5520** | **0.5332** | **0.272** | **0.7509** | **0.6464** | **0.5716** |
| | Fuzzy-Cmeans | 0.1954 | 0.282 | 0.4471 | 0.2707 | 0.2658 | 0.138 | 0.4910 | 0.5020 | 0.3240 |
| Qwen 30x2.7B | HC-SMoE | **0.3532** | **0.6149** | **0.7535** | **0.4695** | **0.4534** | **0.228** | **0.6606** | **0.6456** | **0.5223** |
| | Fuzzy-Cmeans | 0.1954 | 0.2816 | 0.4428 | 0.2708 | 0.2655 | 0.136 | 0.4946 | 0.5043 | 0.3239 |

performance often falling below random-guess baselines. In contrast, HC-SMoE maintains competitive accuracy even at a 75% reduction through its output-based clustering and merging strategy. This finding can be attributed to the fact that experts with similar outputs are likely to capture related features or patterns in the data. Merging these experts allows the model to preserve essential information while reducing redundancy, which enables a more compact representation without compromising performance.

## C. Efficiency Discussion

We evaluate computational and memory costs on the Mixtral 8x7B and Qwen1.5-MoE-A2.7B-Chat models in both their original and merged versions. All experiments use the same calibration dataset as the main experiments and consist of 32 sequences of 2048 tokens sampled from the C4 corpus (Raffel et al., 2020). The results in Table 20 show that a reduction in the number of experts leads to significant decreases in memory usage and GLOPs without impact on throughput and latency. The ideal benefits of reduced router latency from fewer output channels are not realized since we retain the original router weights to prevent accuracy degradation. As a result, the router functions as if the original number of experts exists, with experts within the same group producing identical outputs through their corresponding merged experts.

In addition to the inference costs, we also compared the runtime and memory usage of HC-SMoE algorithm against various baselines in Table 21 and Table 22. The results demonstrate that HC-SMoE achieves competitive runtime and memory efficiency across different models while maintaining superior performance on benchmarks.

## D. Cluster Quality Analysis

In this section, we analyze the characteristics and measure the cluster quality of K-Means and hierarchical clustering to shed a light on the motivation of using hierarchical clustering instead of K-Means.

We selected hierarchical clustering over K-means based on two primary reasons: its stability and determinism. The initialization of cluster centroids in K-means is often random, which can result in different clustering outcomes across

*Table 17.* Comparison of HC-SMoE and Fuzzy-Cmeans on Mixtral 8x7B-v0.1.

| Model | Method | ARC-c | ARC-e | BoolQ | HellaSwag | MMLU | OBQA | RTE | Winogrande | Average |
|-------|--------|-------|-------|-------|-----------|------|------|-----|------------|---------|
| Mixtral 8x7B | - | 0.5648 | 0.8422 | 0.8505 | 0.6490 | 0.6712 | 0.350 | 0.7112 | 0.7593 | 0.6748 |
| Mixtral 6x7B | HC-SMoE | **0.5145** | **0.8043** | **0.8554** | **0.6142** | **0.6043** | **0.324** | **0.6715** | **0.7514** | **0.6425** |
| | Fuzzy-Cmeans | 0.4804 | 0.7694 | 0.8297 | 0.5953 | 0.5282 | 0.308 | 0.639 | 0.7427 | 0.6116 |
| Mixtral 4x7B | HC-SMoE | **0.4573** | **0.7454** | **0.8018** | **0.5709** | **0.4571** | **0.270** | **0.5523** | **0.7285** | **0.5729** |
| | Fuzzy-Cmeans | 0.3609 | 0.6456 | 0.7339 | 0.4704 | 0.3725 | 0.240 | 0.5343 | 0.6654 | 0.5029 |

*Table 18.* Zero-shot performance evaluation of HC-SMoE and three baseline methods on Qwen1.5-MoE-A2.7B-Chat with expert reduction to 23 and 15 per layer. We exclude O-prune for this experiment due to the large search space.

| Model | Method | ARC-c | ARC-e | BoolQ | HellaSwag | MMLU | OBQA | RTE | Winogrande | Average |
|-------|--------|-------|-------|-------|-----------|------|------|-----|------------|---------|
| Qwen 60x2.7B | None | 0.3951 | 0.7012 | 0.8135 | 0.5932 | 0.6047 | 0.310 | 0.7329 | 0.6559 | 0.6008 |
| Qwen 23x2.7B | F-prune | 0.2287 | 0.3763 | 0.5957 | 0.3627 | 0.2413 | 0.186 | 0.5668 | 0.5280 | 0.3857 |
| | S-prune | 0.2150 | 0.4200 | 0.5945 | 0.3307 | 0.2725 | 0.166 | 0.5343 | 0.5406 | 0.3842 |
| | MC-SMoE | 0.2014 | 0.2803 | 0.4410 | 0.2743 | 0.2292 | 0.158 | 0.4982 | 0.5114 | 0.3242 |
| | HC-SMoE (ours) | **0.3319** | **0.5720** | **0.7554** | **0.4111** | **0.3957** | **0.216** | **0.6606** | **0.6117** | **0.4943** |
| Qwen 15x2.7B | F-prune | 0.2176 | 0.3026 | 0.5269 | 0.2871 | 0.2358 | 0.154 | 0.5018 | 0.5185 | 0.3430 |
| | S-prune | 0.1954 | 0.3114 | 0.5275 | 0.2803 | 0.2537 | 0.136 | 0.5199 | 0.5122 | 0.3421 |
| | MC-SMoE | 0.1903 | 0.3035 | 0.3966 | 0.2741 | 0.2295 | 0.160 | 0.5199 | 0.5028 | 0.3221 |
| | HC-SMoE (ours) | **0.2662** | **0.5034** | **0.7046** | **0.3664** | **0.3629** | **0.196** | **0.6173** | **0.5777** | **0.4493** |

multiple runs on the same dataset (Ikotun et al., 2023). This non-deterministic behavior of K-means makes it less suitable for tasks where the reproducibility and consistency of clustering results are crucial, such as in the evaluation of downstream tasks. In contrast, hierarchical clustering generates deterministic results for a given dataset and linkage method. This deterministic property ensures that the clustering results are reproducible and consistent across different runs. Furthermore, hierarchical clustering employs a systematic approach to merge clusters based on a specified linkage criterion. This linkage criterion determines the distance between clusters and governs the merging process. By optimizing the linkage criterion, hierarchical clustering guarantees the formation of stable clusters that minimize the intra-cluster distance and maximize the inter-cluster distance. This optimization ensures that the resulting clusters are compact and well-separated, which is desirable for effective pruning and merging of experts in the context of model compression.

In addition to the theoretical justification, we further validated the effectiveness of hierarchical clustering compared to K-means through experiments. We conducted experiments using both methods across three similarity metrics (i.e., router logits, expert outputs, and expert weights) and evaluated six clustering criteria:

1. **L2 distance**: $||T(x) - S(x)||_2$, where $T(x)$ and $S(x)$ represent the outputs of the original and pruned models, respectively. Lower values are better.

2. **Cosine similarity**: cosine-similarity$(T(x), S(x))$. Higher values are better.

3. **Silhouette score (Euclidean)**: Measures how similar an object is to its cluster compared to other clusters, using Euclidean distance. Higher values are better.

4. **Dunn index (Euclidean)**: Evaluates cluster compactness and separation, using Euclidean distance. Higher values are better.

5. **Silhouette score (Cosine)**: Similar to (3) but based on cosine similarity. Higher values are better.

6. **Dunn index (Cosine)**: Similar to (4) but based on cosine similarity. Higher values are better.

*Table 19.* Zero-shot performance evaluation of HC-SMoE and four baseline methods on Mixtral 8x7B-v0.1 with expert reduction to three and two per layer. The runtime for each algorithm is provided in seconds.

| Model | Method | ARC-c | ARC-e | BoolQ | HellaSwag | MMLU | OBQA | RTE | Winogrande | Average | Time (s) |
|---|---|---|---|---|---|---|---|---|---|---|---|
| Mixtral 8x7B | None | 0.5648 | 0.8422 | 0.8505 | 0.6490 | 0.6712 | 0.350 | 0.7112 | 0.7593 | 0.6748 | - |
| Mixtral 3x7B | F-prune | 0.2253 | 0.399 | 0.6024 | 0.3663 | 0.2414 | 0.168 | 0.5379 | 0.5249 | 0.3832 | 61.070 |
| | S-prune | 0.2082 | 0.3826 | 0.5951 | 0.3648 | 0.2315 | 0.154 | 0.509 | 0.5383 | 0.3729 | 54.000 |
| | O-prune | **0.4471** | **0.7210** | **0.7761** | 0.5377 | 0.3847 | 0.264 | **0.5921** | **0.7024** | **0.5531** | 2530.584 |
| | MC-SMoE | 0.2125 | 0.2963 | 0.6131 | 0.2699 | 0.2513 | 0.126 | 0.5162 | 0.5185 | 0.3505 | 43.544 |
| | HC-SMoE (ours) | 0.4078 | 0.7138 | 0.7755 | **0.5402** | **0.4156** | **0.268** | 0.5451 | 0.7001 | 0.5458 | 253.299 |
| Mixtral 2x7B | F-prune | 0.2329 | 0.2689 | 0.6214 | 0.2681 | 0.2574 | 0.150 | 0.491 | 0.5162 | 0.3507 | 61.868 |
| | S-prune | 0.2022 | 0.2929 | 0.6193 | 0.2942 | 0.2356 | 0.142 | 0.5199 | 0.5083 | 0.3518 | 55.718 |
| | O-prune | 0.3481 | 0.6540 | 0.7043 | **0.4846** | 0.3163 | 0.214 | **0.5451** | **0.6685** | 0.4919 | 1181.15 |
| | MC-SMoE | 0.2116 | 0.2908 | 0.6196 | 0.2697 | 0.2370 | 0.132 | 0.4729 | 0.5107 | 0.3430 | 43.778 |
| | HC-SMoE (ours) | **0.3746** | **0.6721** | **0.7541** | 0.4786 | **0.3606** | **0.236** | 0.5307 | 0.6582 | **0.5081** | 267.134 |

*Table 20.* Evaluation of computational and memory efficiency across multiple models. For Mixtral: Mixtral 8x7B (original), Mixtral 6x7B (25% pruned), and Mixtral 4x7B (50% pruned). For Qwen1.5-MoE-A2.7B-Chat: Qwen 60x2.7B (original), Qwen 45x2.7B (25% pruned), and Qwen 30x2.7B (50% pruned). All measurements use identical input sequences and include throughput (tokens per ms), latency (s), GFLOPs, model memory, and model size (number of parameters).

| Models | Throughput | Latency | GFLOPs | Memory | Model Size |
|---|---|---|---|---|---|
| Mixtral 8x7B | $13.45 \pm 1.30$ | $2.854 \pm 0.333$ | 2989 | 87.49GB | 46.7B |
| Mixtral 6x7B | $13.87 \pm 0.47$ | $2.666 \pm 0.093$ | 2267 | 66.49GB | 35.4B |
| Mixtral 4x7B | $13.96 \pm 0.65$ | $2.599 \pm 0.166$ | 1546 | 45.49GB | 24.2B |
| Qwen 60x2.7B | $24.08 \pm 0.17$ | $1.593 \pm 0.168$ | 916 | 27.04GB | 14.3B |
| Qwen 45x2.7B | $23.95 \pm 0.24$ | $1.541 \pm 0.011$ | 717 | 21.23GB | 11.2B |
| Qwen 30x2.7B | $23.16 \pm 0.42$ | $1.583 \pm 0.034$ | 518 | 15.44GB | 8.1B |

Silhouette score evaluates clustering quality at the data point level ( i.e., expert level), while the Dunn index evaluates it at the cluster level. The Dunn index considers maximum intra-cluster and minimum inter-cluster distances, while the Silhouette score uses mean distances. Both metrics highlight clustering compactness and separability. We excluded evaluations involving the cosine similarity of expert weights due to the high computational cost of processing concatenated weight tensors, which would require excessive GPU memory. The detailed formulation is provided at the bottom of this response.

Table 23 summarizes the results. Hierarchical clustering with expert outputs achieves the lowest L2 error and the highest cosine similarity with the original model outputs at 25% and 50% pruning ratios. Moreover, hierarchical clustering consistently outperforms K-means across most clustering metrics, demonstrating better clustering quality. These results substantiate the stability and effectiveness of hierarchical clustering in producing compact, well-separated clusters. Furthermore, the zero-shot performance on eight language tasks, as demonstrated in Table 5 in our manuscript, further supports the superiority of hierarchical clustering. Across all tasks, hierarchical clustering consistently outperforms K-means, and achieves better and more stable accuracy.

Based on the above reasons, hierarchical clustering is preferable for its deterministic nature, superior clustering quality, and consistent performance across similarity metrics and benchmarks.

# E. Frequency Analysis

### E.1. Mixtral 8x7B

We present the activation frequency analysis of all experts in Mixtral 8x7B (Jiang et al., 2024) using our sampling dataset from C4 (Raffel et al., 2020) and eight language benchmarks. The results provide evidence against using frequency as the sole criterion for determining the number of experts in each layer. The analysis reveals variability in activation frequency

*Table 21.* Runtime and memory consumption of HC-SMoE and baseline of Mixtral8x7B on 8 NVIDIA V100 GPUs.

| Model | Method | Runtime (sec) | Memory (GB) |
|---|---|---|---|
| Mixtral 6x7B | F-prune | 65 | 106.600 |
| | S-prune | 63 | 106.592 |
| | O-prune | 1891 | 122.640 |
| | M-SMoE | 47 | 106.601 |
| | HC-SMoE (ours) | 111 | 138.039 |
| Mixtral 4x7B | F-prune | 63 | 106.6005 |
| | S-prune | 64 | 106.5926 |
| | O-prune | 3605 | 122.640 |
| | M-SMoE | 50 | 106.601 |
| | HC-SMoE (ours) | 112 | 138.039 |

*Table 22.* Runtime and memory consumption of HC-SMoE and baseline of Qwen1.5-MoE-A2.7B-Chat on 4 NVIDIA V100 GPUs. Note that since O-prune has non-feasible computation time on Qwen model, we only run 100 iterations for each layer.

| Model | Method | Runtime (sec) | Memory (GB) |
|---|---|---|---|
| Qwen 45x2.7B | F-prune | 95 | 61.605 |
| | S-prune | 63 | 106.592 |
| | O-prune (100) | 824 | 70.849 |
| | M-SMoE | 107 | 48.829 |
| | HC-SMoE (ours) | 290 | 48.701 |
| Qwen 30x2.7B | F-prune | 95 | 61.605 |
| | S-prune | 95 | 61.605 |
| | O-prune | 840 | 70.849 |
| | M-SMoE | 107 | 48.829 |
| | HC-SMoE (ours) | 323 | 48.701 |

across different tasks, highlighting the fact that this metric is not a consistent or reliable indicator for expert selection in task-agnostic settings

### E.2. TinyLLama-4x1.1B-MoE

The activation frequency analysis of all experts in TinyLLaMa-4x1.1B-MoE [2] on our sampling dataset of C4 (Raffel et al., 2020) and eight language benchmarks. It can be the evidence of poor expert utilization in SMoE, since one of the experts is seldom chosen among all tasks.

---

[2] https://huggingface.co/s3nh/TinyLLama-4x1.1B-MoE

*Table 23.* The ablation study of measuring the error of the last layer with the original model, and the cluster quality with different clustering methods and similarity metrics on the Qwen model. We conduct the measurement on Qwen 45x2.7B (pruning ratio 25%) and Qwen 30x2.7B (pruning ratio 50%). We bold the score of highest Silhouette score and Dunn Index every two row because these two criteria cannot directly compared to methods using different metrics, i.e., methods using expert outputs cannot compare with methods using weights.

| Model | Cluster | Metric | L2 error | Cosine Similarity | Silhouette-Euc | Dunn-Euc | Silhouette-Cos | Dunn-Cos |
|---|---|---|---|---|---|---|---|---|
| | HC | *eo* | **3,806.8332** | **0.9972** | **0.7909** | **0.8252** | **0.7090** | 0.5136 |
| | Kmeans | *eo* | 6,769.3674 | 0.9910 | 0.6093 | 0.2145 | 0.6489 | 0.5300 |
| | HC | *weight* | 9,307.8344 | 0.9874 | **0.7358** | **0.9801** | - | - |
| Qwen 45x2.7B | Kmeans | *weight* | 7,962.1627 | 0.9919 | 0.6125 | 0.9225 | - | - |
| | HC | *rl* | 7,572.2246 | 0.9897 | **0.6688** | **0.8453** | **0.7169** | **0.5204** |
| | Kmeans | *rl* | 7,463.5265 | 0.9900 | 0.6196 | 0.4810 | 0.6389 | 0.3469 |
| | HC | *eo* | **8,142.6961** | **0.9878** | **0.6104** | **0.8126** | **0.4233** | 0.4939 |
| | Kmeans | *eo* | 9,796.4269 | 0.9842 | 0.1851 | 0.2210 | 0.2375 | **0.5136** |
| | HC | *weight* | 11,400.2119 | 0.9791 | **0.4876** | **0.9734** | - | - |
| Qwen 30x2.7B | Kmeans | *weight* | 10,894.4059 | 0.9808 | 0.2380 | 0.9245 | - | - |
| | HC | *rl* | 10,022.4158 | 0.9826 | **0.4495** | **0.5536** | **0.4673** | **0.2115** |
| | Kmeans | *rl* | 10,348.0981 | 0.9825 | 0.2566 | 0.3731 | 0.2653 | 0.2030 |

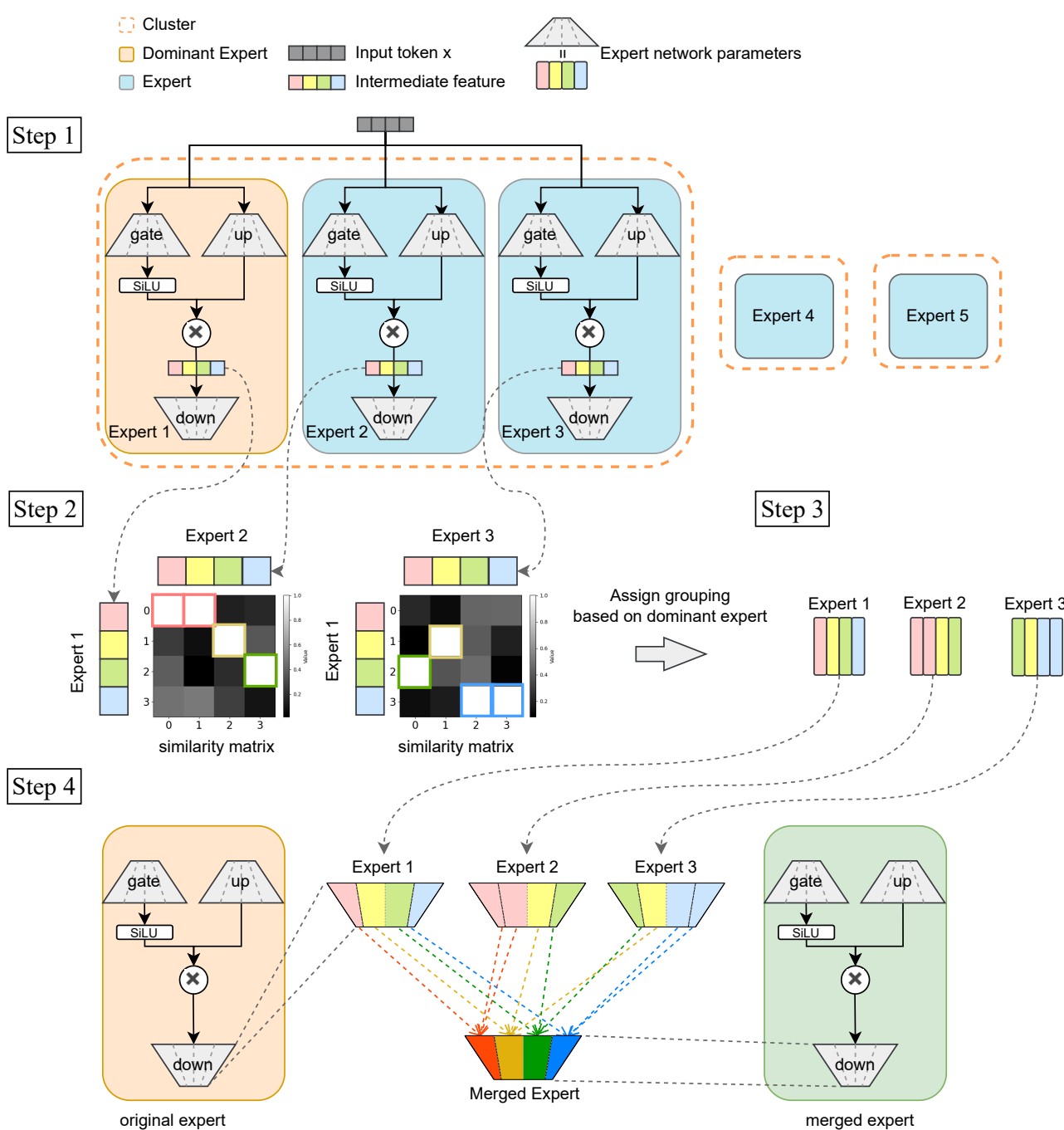

Figure 4. Fix-dominant merging. Given experts within cluster and dominant expert index, *Step 1.* we first collect intermediate features from each experts. *Step 2.* Then, we use pairwise correlation to compare similarity between dominant expert and non-dominant experts. *Step 3.* Each non-dominant expert's dimension choose the dimension of highest similarity with its in the dominant expert feature as group. *Step 4.* Based on this grouping, we average merge each expert weights in each dimension.

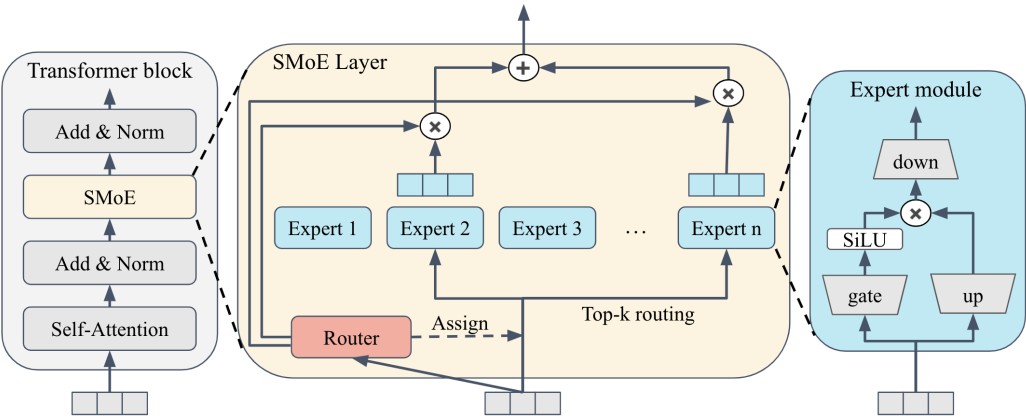

*Figure 5.* General architecture of SMoE. The router uses top-2 routing to assign each token to the two experts with the highest scores.

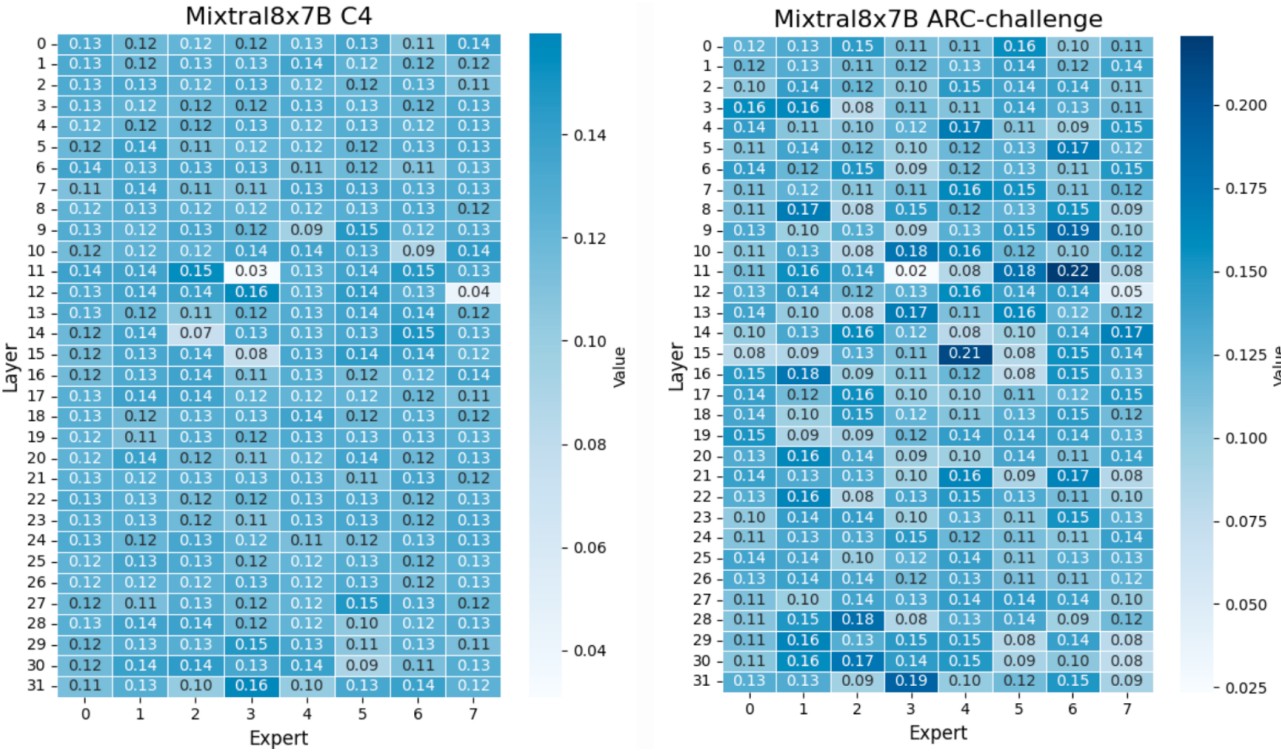

*Figure 6.* The frequency anslysis of Mixtral 8x7B on ARC-c and our sampling dataset of C4.

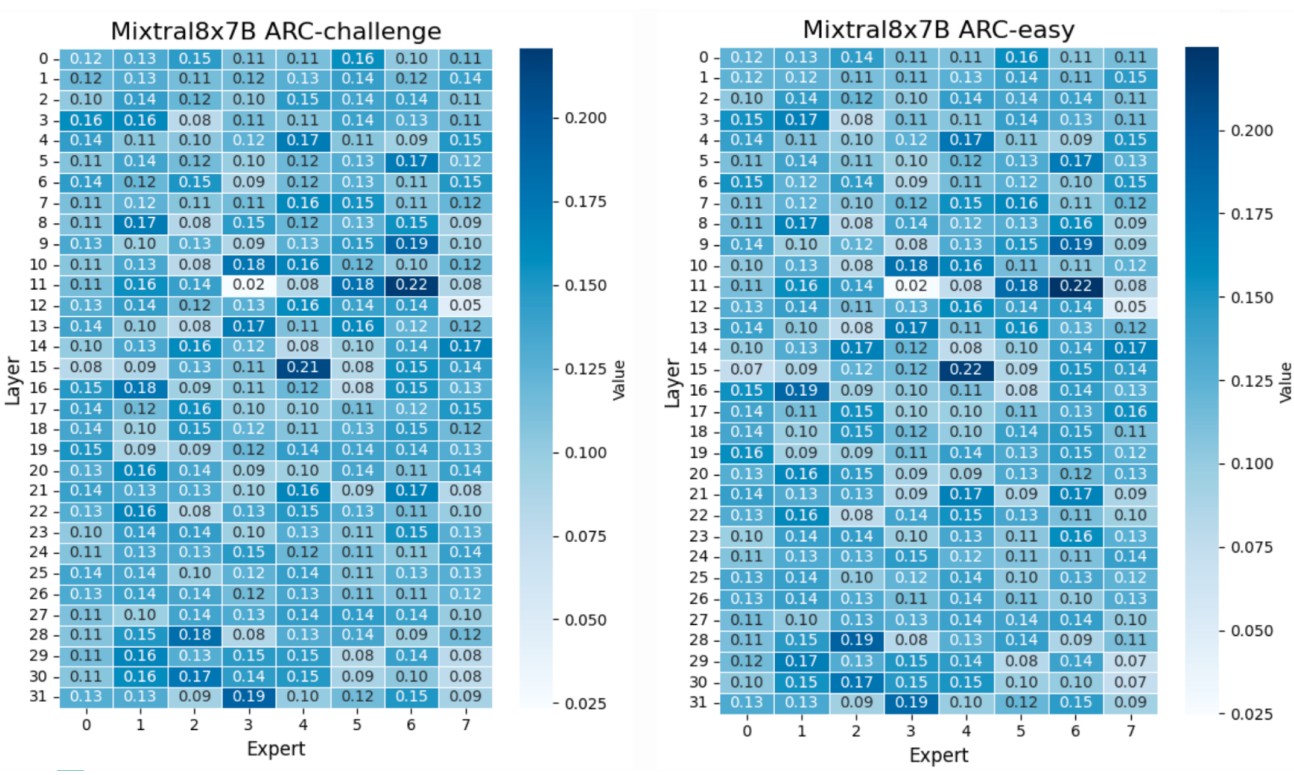

*Figure 7.* The frequency anslysis of Mixtral 8x7B on ARC-c and ARC-e.

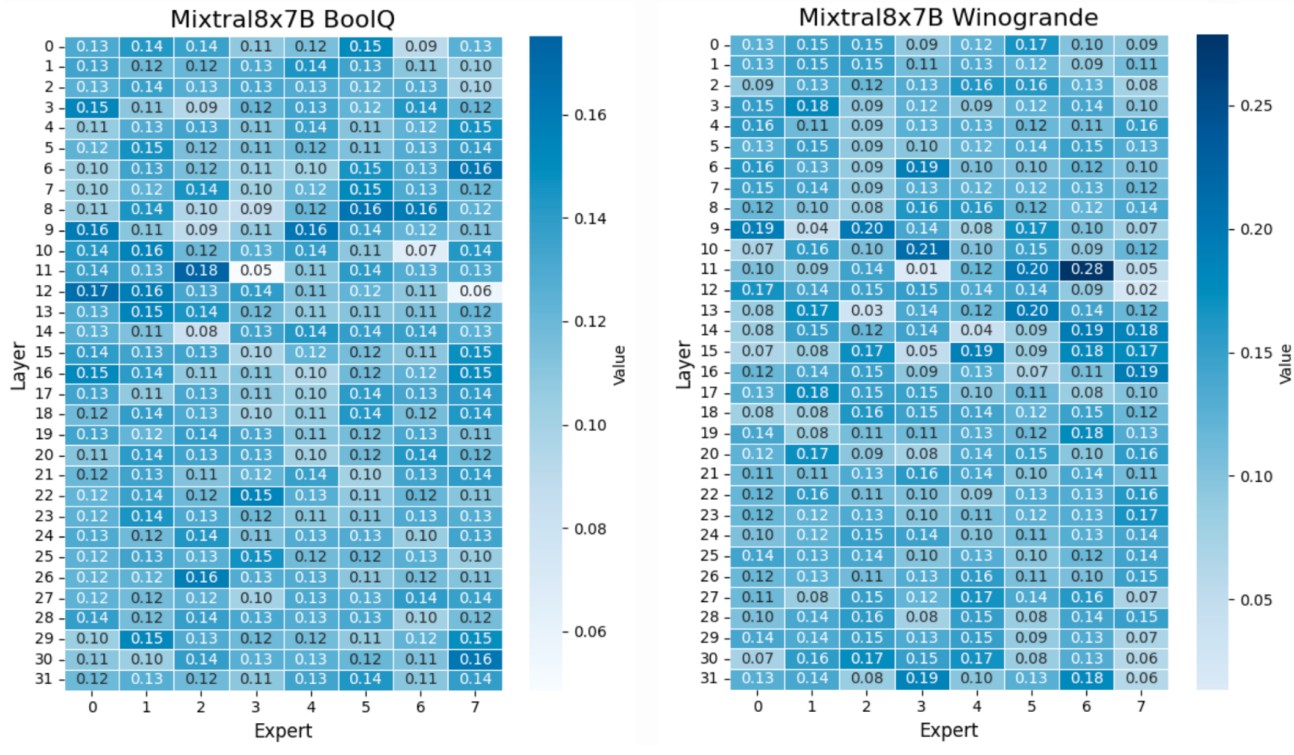

*Figure 8.* The frequency anslysis of Mixtral 8x7B on BoolQ and Winogrande.

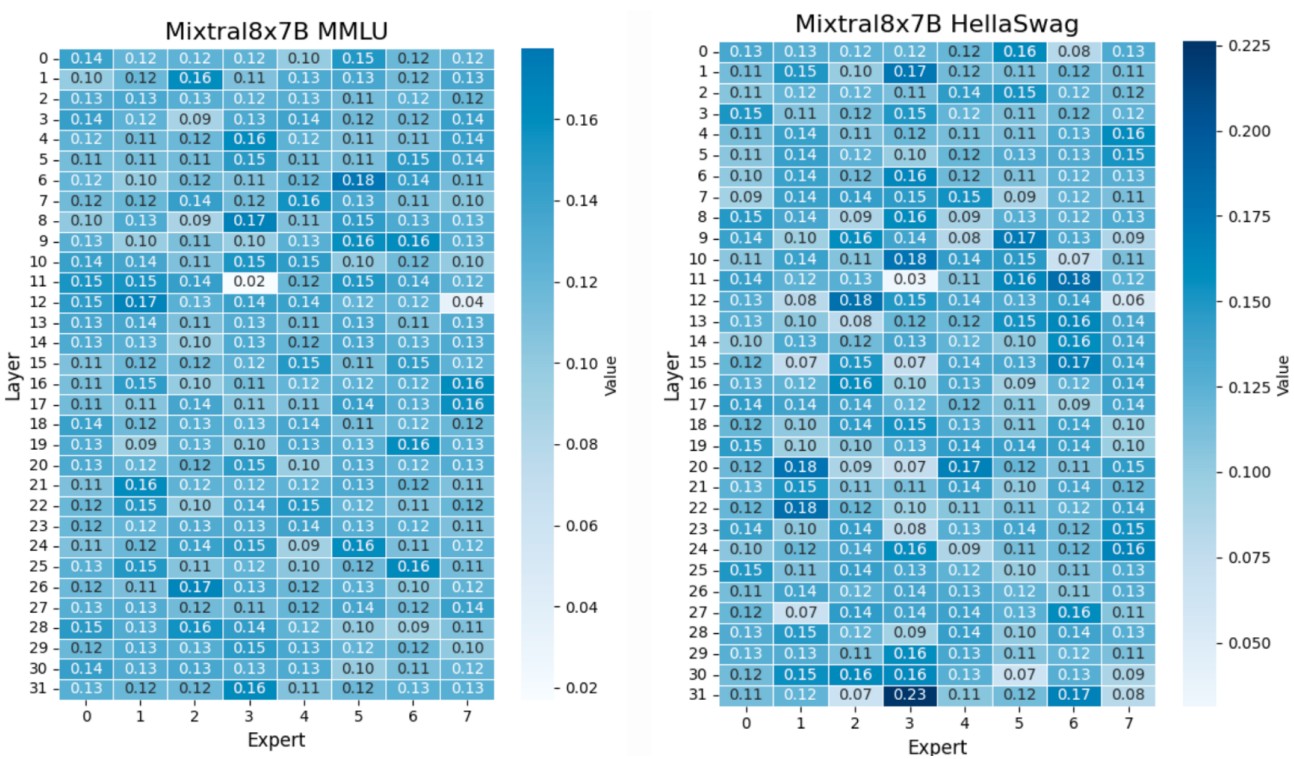

*Figure 9.* The frequency anslysis of Mixtral 8x7B on MMLU and HellaSwag.

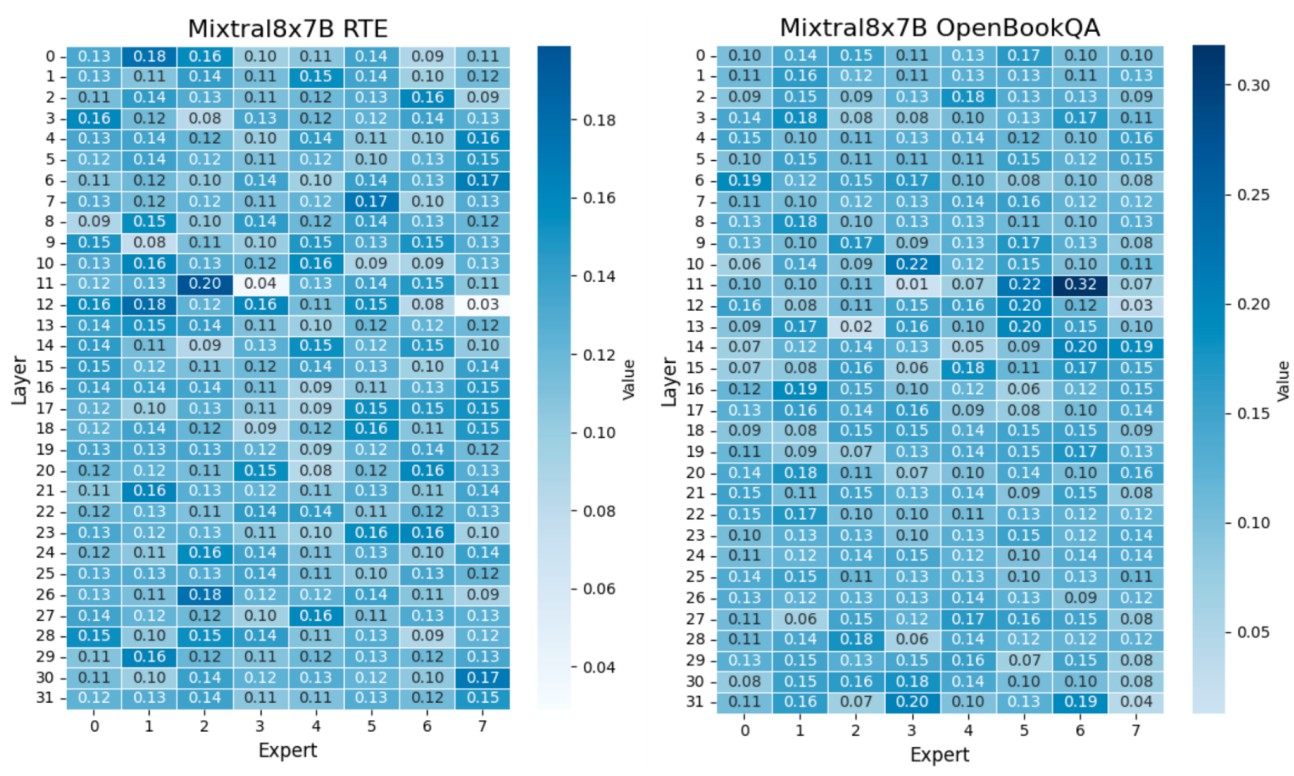

*Figure 10.* The frequency anslysis of Mixtral 8x7B on RTE and OpenBookQA.

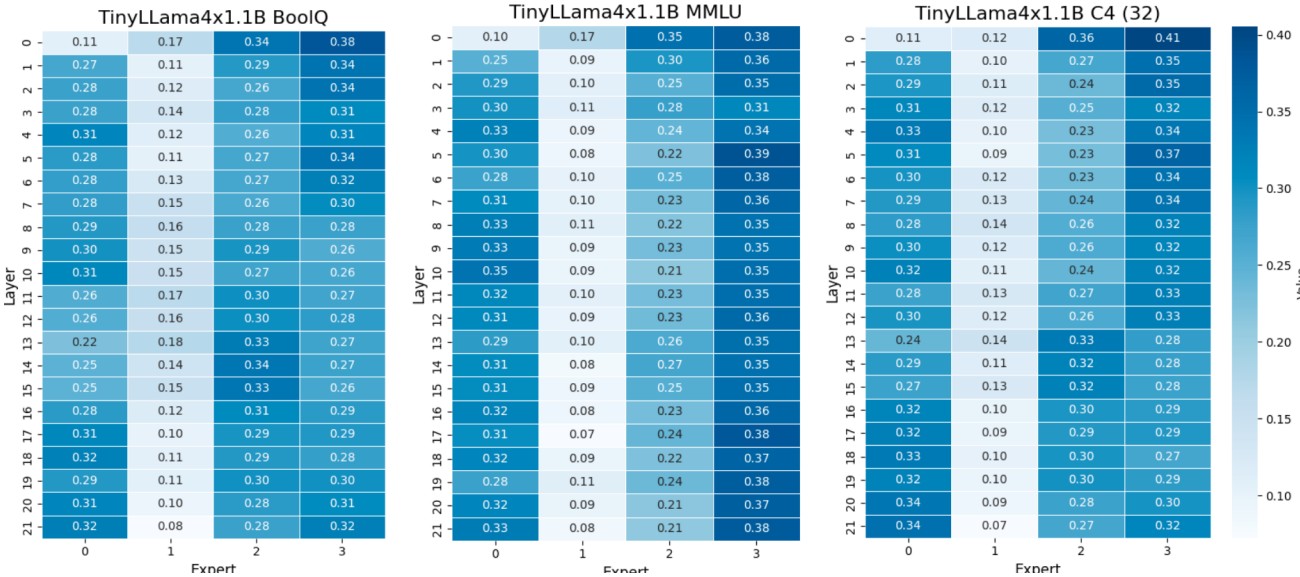

*Figure 11.* The frequency anslysis of TinyLLaMa-4x1.1B-MoE on BoolQ, MMLU and sampling dataset of C4.

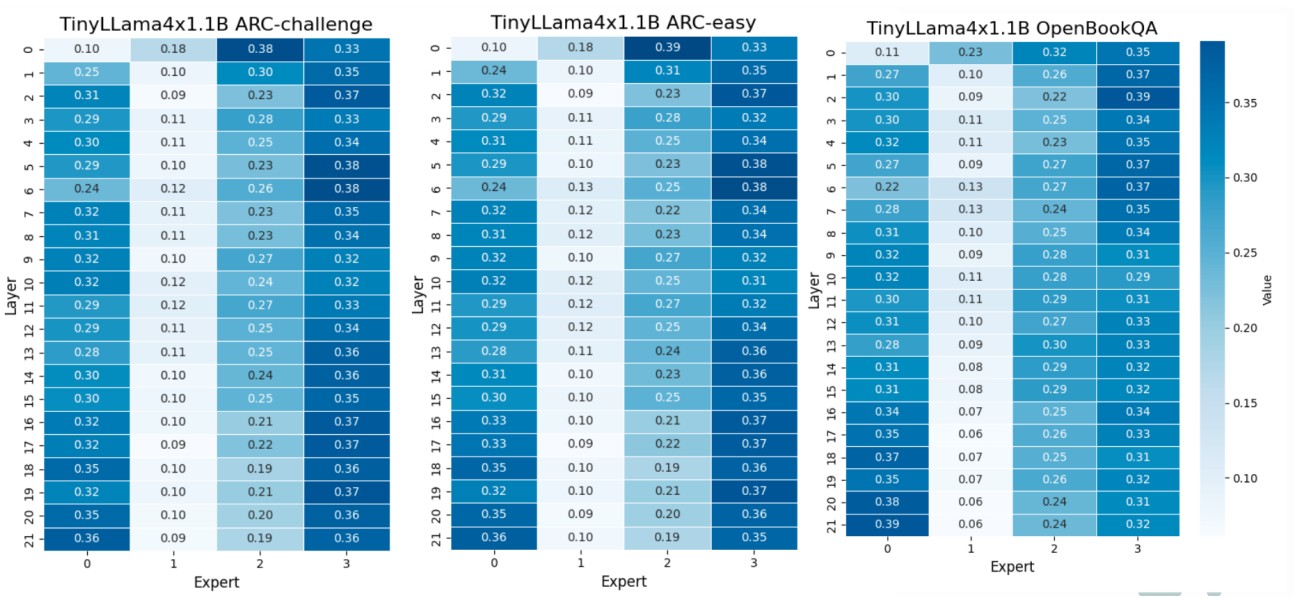

*Figure 12.* The frequency anslysis of TinyLLaMa-4x1.1B-MoE on ARC-c, ARC-e and OpenBookQA.

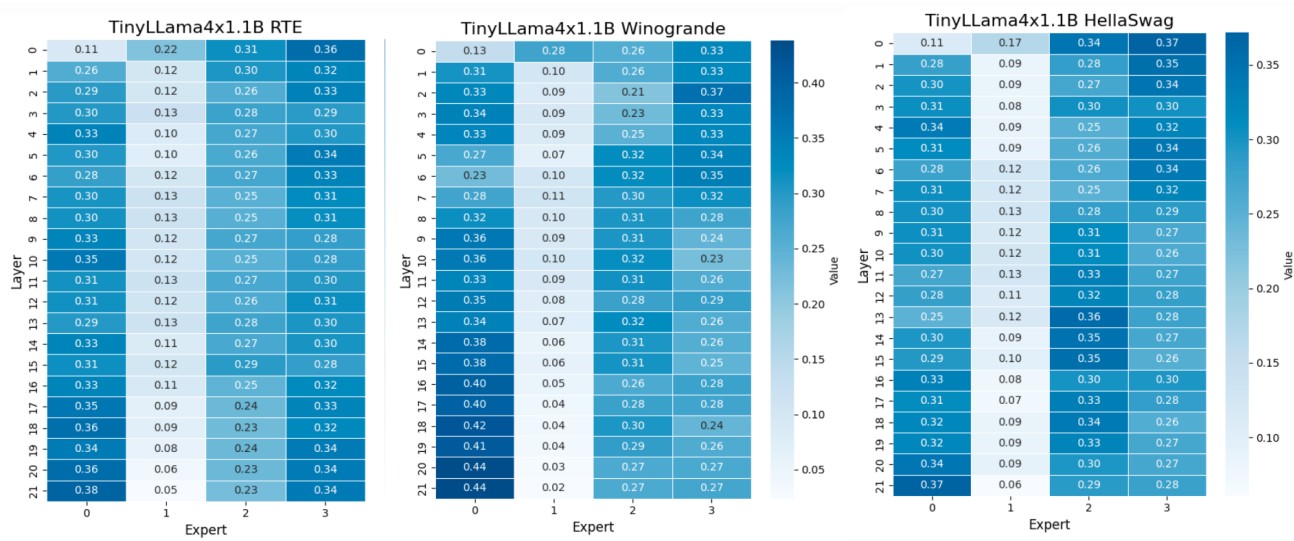

*Figure 13.* The frequency anslysis of TinyLLaMa-4x1.1B-MoE on RTE, Winogrand'e and HellaSwag.

