# OpenReview forum: "Retraining-free Merging of Sparse MoE via Hierarchical Clustering"
_ICML.cc/2025/Conference — ICML 2025 poster_

### Official Review · Reviewer_cVaE · 2025-03-10

**Overall Recommendation:** 1

**Summary:**

This paper introduces HC-SMoE, a retraining-free framework for merging experts in Sparsely Activated Mixture-of-Experts (SMoE) models via hierarchical clustering. The key idea is to group experts based on their output similarities over a calibration dataset, followed by frequency-weighted merging to reduce model parameters while preserving performance. The authors validate HC-SMoE on Qwen and Mixtral models, demonstrating superior performance over pruning baselines (e.g., O-prune, S-prune) and merging methods (e.g., M-SMoE) across multiple zero-shot tasks. The main contributions include: (1) output-based similarity metrics for clustering, (2) hierarchical clustering for improved robustness, and (3) empirical validation across diverse benchmarks.

**Claims And Evidence:**

The central claim—that HC-SMoE outperforms existing pruning/merging methods—is are supported by extensive experiments (Tables 2–3, 6–7). However, theoretical justification for why hierarchical clustering is optimal is lacking.

**Essential References Not Discussed:**

Key Contributions vs. Literature:
1.	The work builds on SMoE compression methods like TSEP (Chen et al., 2022), O-prune (Lu et al., 2024), and M-SMoE (Li et al., 2024). The novelty lies in hierarchical clustering for merging, contrasting with prior pruning or single-pass grouping.
Missing Citations:
1.	Cluster-based routing MoE has been proposed in many papers before(e.g., "On the Benefits of Learning to Route in Mixture-of-Experts Models" and "Once Read is Enough: Domain-specific Pretraining-free Language Models with Cluster-guided Sparse Experts for Long-tail Domain Knowledge"), but it is not proposed and referenced in this paper. Such papers have a more detailed analysis of the clustering phenomenon in the representation space, but in this paper, it is missing.

**Ethical Review Concerns:**

As the impact statement was not provided in the article as required, I cannot comment on this.

**Experimental Designs Or Analyses:**

Experiments are comprehensive but lack latency/throughput comparisons (Table 19 only reports FLOPs and memory). Inference efficiency gains from merging are unclear.

**Methods And Evaluation Criteria:**

The article lacks rigorous analysis and mathematical modeling of the rationality of the method, and the article mainly provides empirical explanations.

**Other Comments Or Suggestions:**

N/A

**Other Strengths And Weaknesses:**

Strengths:
1.	Practical Impact: HC-SMoE offers a deployable solution for resource-constrained settings.
2.	Scalability: Validated on large models (Mixtral 8x7B) with significant parameter reduction.
Weaknesses:
1.	Theoretical Gaps: No formal analysis of clustering quality or merging stability.
2.	Semantic Preservation: Claims about preserving semantic spaces are unsubstantiated (see Questions).
3.	Ethical Statement: Missing required impact statement on societal/environmental implications.
Clarity:
1.	The method is well-described, but Figure 2 (clustering illustration) is too simplistic.

**Questions For Authors:**

The proposed clustering method is multi-level, but it takes into account two premises: a. Many papers pointed out that experts of token-level MoE did not have a preference for actual professional semantics(e.g., "DeepSeekMoE: Towards Ultimate Expert Specialization in Mixture-of-Experts Language Models" and "OpenMoE: An Early Effort on Open Mixture-of-Experts Language Models"). b. The professional semantic preferences of different experts are also not clearly indicated in this article. Then the questions that authors should explain are:
a. How does HC-MOE guarantee that experts merging based on Hierarchical Clustering will not produce semantic conflicts?
b. Does HC-MoE cause a huge impact on the semantic space of the original model? It is a pity that neither of these issues is explicitly explained in the text.

**Relation To Broader Scientific Literature:**

N/A

**Theoretical Claims:**

No theoretical proofs are provided. For example, the claim that hierarchical clustering "produces theoretically guaranteed groupings" (Section 1) is unsupported. A theoretical analysis of clustering robustness or error bounds is missing.

---

> ### Author Rebuttal · Authors · 2025-04-01
>
> We appreciate the reviewer’s valuable feedback and effort spent on the review, and would like to respond to the reviewer’s questions as follows.
>
> **Q1.**  Theoretical Gaps: No formal analysis of clustering quality or merging stability.
>
> **Response:**
> We appreciate the reviewer's suggestion regarding theoretical analysis. We would like to direct the reviewer to refer to the general response section. Please refer to the theoretical justification link below.
> - [theoretical justification](https://anonymous.4open.science/r/ICML_Rebuttal-0632/Theoretical_Justification.md)
>
> ---
>
> **Q2.** Experiments are comprehensive but lack latency/throughput comparisons (Table 19 only reports FLOPs and memory). Inference efficiency gains from merging are unclear.
>
> **Response:**
> We thank the reviewer for this observation. Our primary objective focuses on reducing parameter count in MoE models without retraining while maintaining computational efficiency. Table 19 demonstrates significant reductions in FLOPs and memory usage, which highlights the efficiency gains achieved through our approach.
>
> It is important to note that our method preserves the top-K routing mechanism, wherein each token continues to be assigned to $K$ experts per MoE layer, the same as the original model. As a result, when we reduce the number of experts in each layer to $r$ while ensuring $r \ge K$, the inference cost remains equivalent to that of the original model. While merging reduces storage and computational requirements per expert, the routing mechanism dictates that inference cost per token primarily depends on K rather than the total number of experts.
>
>
> ---
>
> **Q3.** Does HC-MoE cause a huge impact on the semantic space of the original model? It is a pity that neither of these issues is explicitly explained in the text.
>
> **Response:**
> To address concerns regarding potential semantic shifts, we have incorporated t-SNE visualizations that compare expert representations before and after merging. These results demonstrate that HC-SMoE effectively preserves the model's semantic coherence, as the expert distributions maintain substantial consistency after the merging process.
>
> Specifically, we visualize the first MoE layer of Mixtral-8x7B, utilizing the same calibration dataset described in Section 4.1. We collect 65,536 output vectors per expert, each with a hidden dimension of 4,096, by directing identical MoE input tokens to all experts. Each point in the t-SNE plot represents the average of 128 token outputs, which then undergoes projection into two dimensions using sklearn.manifold.TSNE with n_components = 2. The analysis yields several key observations:
> - With perplexity = 8, the t-SNE visualization of the original model reveals a clear eight-cluster structure, which corresponds to the eight experts.
> - After applying HC-SMoE to reduce the expert set to six experts, the resulting t-SNE plot maintains a well-defined six-cluster structure, despite some overlap. This indicates that HC-SMoE maintains expert specialization to a significant extent, and preserves the output distribution of the original model.
>
> It is important to note that t-SNE visualization at the single-token level would appear as random noise without discernible cluster structure. This occurs because individual token embeddings exist in high-dimensional space and do not inherently form clusters without appropriate aggregation.
>
> - [t-SNE of each expert’s output of the original Mixtral8x7B's first layer](https://anonymous.4open.science/r/ICML_Rebuttal-0632/t-SNE/t-SNE-8e_layer_0_output.png).
> - [t-SNE of each expert’s output of first layer after HC-SMoE which reduced each layer in Mixtral8x7B to 6 experts](https://anonymous.4open.science/r/ICML_Rebuttal-0632/t-SNE/t-SNE_6e_layer_0_output.png).
> - [t-SNE of each expert’s output of the original model layer where each point indicates a single output token](https://anonymous.4open.science/r/ICML_Rebuttal-0632/t-SNE/t-SNE_150tokens_for_each_expert_output.png).
>
> ---
>
> Due to space limitations, we can only provide concise responses here. We have more detailed and comprehensive answers regarding the concerns on ethical statement, recommend related works on cluster-based routing MoE as well as [figure 2's enhancement](https://anonymous.4open.science/r/ICML_Rebuttal-0632/figure2_modified.png) and other questions, which we look forward to discussing thoroughly with the reviewer in the next phase of the review process.

---

### Official Review · Reviewer_39Jv · 2025-03-13

**Overall Recommendation:** 3

**Summary:**

The paper presents HC-SMoE, a new framework for reducing SMoE model parameters that doesn't require retraining and works across different tasks. HC-SMoE uses hierarchical clustering on expert outputs and frequency-weighted merging, which offers two main benefits over previous approaches. First, it uses iterative comparisons to create expert groups, leading to better diversity between groups and similarity within groups. Second, it measures similarity based on expert outputs rather than router logits, making it more generalizable across different datasets.

**Claims And Evidence:**

Yes

**Essential References Not Discussed:**

Related works that are essential to understanding the (context for) key contributions of the paper are discussed.

**Experimental Designs Or Analyses:**

The experimental design and analysis looks sound.

**Methods And Evaluation Criteria:**

Yes, the methods, models and datasets make sense.

**Other Comments Or Suggestions:**

N/A

**Other Strengths And Weaknesses:**

Strengths:
- The paper is well written
- The proposal of averaged expert output seems novel and using HC to find cluster seems appropriate.
- The experiments are thorough

Weaknesses:
- It was not entirely clear to me why Li et al. (2024) proposal is one-shot and why HC is iterative. It would have been nice to include an algorithm in the paper. When using HC, are the experts merged at every step when creating the dendogram? Or the experts are merged only at the desired step of say 25% sparsity?

**Questions For Authors:**

See weaknesses.

**Relation To Broader Scientific Literature:**

The paper is contextualized properly in the context of broader scientific literature.

**Theoretical Claims:**

No theoretical analysis

---

> ### Author Rebuttal · Authors · 2025-04-01
>
> We appreciate the reviewer’s valuable feedback and effort spent on the review, and would like to respond to the reviewer’s questions as follows.
>
> **Q1.** It was not entirely clear to me why Li et al. (2024) proposal is one-shot and why HC is iterative.
>
> **Response:**
> The fundamental distinction between these approaches lies in their underlying methodologies. Li et al.'s expert pruning operates through a one-shot mechanism, wherein only the top $r$ experts with the highest routing scores are retained based on a single evaluation of the data. This process involves computing routing scores by averaging results from a forward pass across the entire calibration dataset, followed by sorting experts according to these averaged scores to identify the top rr r candidates. This methodology proceeds in a direct, non-iterative manner.
>
> In contrast, our HC-SMoE method employs an iterative approach. Hierarchical clustering necessitates multiple sequential steps to systematically merge experts into clusters. Each step selects the optimal pair of clusters (or experts) to combine based on minimizing the average intra-cluster distance. This iterative procedure continues until precisely $r$ clusters are established, ensuring that each clustering decision optimizes the process by minimizing clustering error at every iteration.
>
> **Q2.** It would have been nice to include an algorithm in the paper.
>
> **Response:**
> We acknowledge the reviewer's recommendation regarding a more detailed description of the algorithm, and we have provided [the link to algorithm](https://anonymous.4open.science/r/ICML_Rebuttal-0632/algorithm1.png). Specifically, in Algorithm 1, we outline the steps of hierarchical clustering and expert merging in our methodology.
>
> **Q3.** When using HC, are the experts merged at every step when creating the dendrogram? Or the experts are merged only at the desired step of say 25% sparsity?
>
> **Response:**
> To clarify the merging process: The procedure initially involves clustering experts through the construction of a dendrogram, which provides a hierarchical representation of expert relationships. Upon completion of the dendrogram, we proceed to merge the experts at the final stage to achieve the specified sparsity level. This means that experts are not merged at every step of the dendrogram creation. Instead, they are merged only once the final clustering decision is made, based on the desired sparsity or number of clusters.
>
> We hope this explanation adequately addresses the reviewer’s concerns.

---

### Official Review · Reviewer_YKXx · 2025-03-13

**Overall Recommendation:** 3

**Summary:**

This paper proposes a simple and effective task-agnostic method named HC-SMoE to merge the experts in pre-trained mixture-of-expert models. HC-SMoE first obtains expert outputs on a calibration dataset, and then conducts a hierarchical clustering of the experts based on these outputs. The experts inside a cluster are then merged together. Experiments show that HC-SMoE applied on Qwen and Mixtral achieves better performance than the baseline methods in most cases.

**Claims And Evidence:**

* Section 3.2 claims that "effective clustering enables our method to preserve the capabilities of the original model across diverse merging strategies (Section 3.2.3)." However, the experiment results show that the method does have performance loss than the original model in most cases (see Table 2 and Table 3).

* Section 4.3 claims that "Hierarchical clustering exhibits stability due to its deterministic nature. This stability is evidenced by consistent performance across benchmarks and the highest average scores." However, Table 4 shows that the results have a large variance across different settings. For example, on BoolQ, the performance ranges from 0.3792 to 0.7948.

**Essential References Not Discussed:**

EEP [1] also proposes a training-free algorithm to do expert pruning and merging for MoE models. I understand that it is hard to compare every method in the experiments. But at least, the paper should discuss it, given the high relevance and the fact that EEP has appeared online more than 6 months before the ICML submission deadline.

In addition, given the existence of this paper, the paper might need to tone down some of the claims, such as "task-specific expert pruning ... often necessitate extensive finetuning to maintain performance levels" in Section 1.

[1] https://arxiv.org/abs/2407.00945

**Experimental Designs Or Analyses:**

* Section 4.1 mentioned the experiments of checking how the choice of the calibration dataset affects the results: "To further validate the independence of HC-SMoE from the calibration dataset, we construct two additional datasets from MATH (Hendrycks et al., 2021b) and CodeQA (Liu & Wan, 2021). Please refer to our Appendix B.3 for more details." This is an important experiment. I checked Appendix B, but did not find the results of the baseline methods. Please consider adding baseline results to see if HC-SMoE still outperforms the baselines when the calibration dataset and the evaluation dataset have larger distribution differences.

**Methods And Evaluation Criteria:**

* I read Section B.2, but still find it hard to understand the technical details of the Fixed Dominant Merging approach. For example:
  - What is the definition of "dominant" in Figure 4?
  - Line 577 states that "The merging process then applies an appropriate weighting scheme, such as average merging, preserving the dominant expert’s weight feature order while simplifying the merging process." Could you explain in more detail how the merging is done?
  - What do "feature" and the stars mean in Figure 4?

**Other Comments Or Suggestions:**

Typos:

* Line 51: "from (Li et al., 2024)" --> "from Li et al. (2024)"
* Line 115: "F-prune and M-SMoE" --> "F-prune, and M-SMoE"
* Line 142: "Fig.3." --> "Fig. 3."
* Table 2: The best number is not made bold in the last column of the Qwen 30x2.7B rows.
* Line 584: ”feature” --> "feature"

**Other Strengths And Weaknesses:**

Strengths:

* Overall, the paper is well-written. I really appreciated that the paper not only discusses the proposed approach, but also discusses the rationale behind the design choices and why the choices could be better than other alternatives. These discussions provide useful insights to the readers.

**Questions For Authors:**

In addition to the questions mentioned before, I also have the following questions:

* The proposed method requires a calibration dataset. How large is this dataset in the experiments? How sensitive is the performance of the proposed algorithm to the size of this dataset?

* I found that some of the results across different tables are not consistent. For example, the last row of Table 5 does not match the numbers in Table 4.


# Review summary

Overall, the paper is of good quality. However, given that there are too many unclear points as discussed above, I have to give a negative score. That said, I believe it is feasible to address all these questions during the rebuttal. I would appreciate it if the authors can help clarify these questions, and I would be happy to adjust the score accordingly.

# Reply to "Reply Rebuttal Comment by Authors"

Thank the authors for the further clarification. The numbers in the updated Table 5 still do not match the ones in Table 4. Is it because some settings (e.g., the model) are different? Please make it clearer in the revision.

Since most of my concerns are addressed, I increase the score from 2 to 3.

**Relation To Broader Scientific Literature:**

The key difference to prior work is (1) the use of hierarchical clustering to find the expert merging sets, and (2) the expert similarity is measured based on expert outputs. These are simple changes, and the results show clear improvements.

**Theoretical Claims:**

The paper does not have theoretical claims.

---

> ### Author Rebuttal · Authors · 2025-04-01
>
> We appreciate the reviewer’s valuable feedback and effort spent on the review, and would like to respond to the reviewer’s questions as follows.
>
> ---
>
> **Q1.** Please consider adding baseline results to see if HC-SMoE still outperforms the baselines when the calibration dataset and the evaluation dataset have larger distribution differences.
>
> **Response:**
> We acknowledge the reviewer's valuable inquiry regarding HC-SMoE's performance under distribution shifts. To evaluate the robustness of HC-SMoE against calibration dataset distribution shifts, we conducted additional experiments with MATH and CodeQA as calibration datasets. The results demonstrate the following:
> - HC-SMoE consistently **maintains superior or equivalent performance compared to all baselines** across all configurations on Qwen1.5-MoE-A2.7B-Chat, which demonstrates its capacity to generalize effectively across diverse calibration distributions.
> - When implementing 25% pruning with MATH calibration on Mixtral8x7B, S-prune exhibits marginally superior performance compared to HC-SMoE. However, S-prune demonstrates significantly inferior performance in all other experimental scenarios.
> - F-prune exhibits substantial performance degradation when utilizing MATH and CodeQA, which indicates that **pruning methodologies based solely on frequency or routing scores lack stability**, whereas HC-SMoE maintains robust performance regardless of the calibration dataset employed.
>
> The experimental results with calibration datasets MATH and CodeQA are presented as follows. The red color indicates performance decreases relative to calibration dataset C4, while the green color signifies performance improvements. We excluded O-prune from these experiments as its original publication [4] provides comprehensive analysis regarding the impact of calibration datasets.
>
> - [Qwen on MATH](https://anonymous.4open.science/r/ICML_Rebuttal-0632/Tables/ablation_calib_math_on_qwen)
> - [Qwen on CodeQA](https://anonymous.4open.science/r/ICML_Rebuttal-0632/Tables/ablation_calib_codeqa_on_qwen)
> - [Mixtral on MATH](https://anonymous.4open.science/r/ICML_Rebuttal-0632/Tables/ablation_calib_math_on_mixtral8x7B)
> - [Mixtral on CodeQA](https://anonymous.4open.science/r/ICML_Rebuttal-0632/Tables/ablation_calib_codeqa_on_mixtral8x7B)
>
> ---
>
> **Q2.** The proposed method requires a calibration dataset. How large is this dataset in the experiments? How sensitive is the performance of the proposed algorithm to the size of this dataset?
>
> **Response:**
> Section 4.1 of our original manuscript provides comprehensive details regarding the calibration dataset size. To further evaluate the sensitivity of our method to dataset size, we conducted an additional experiment using Qwen with 50% expert parameter pruning while varying the calibration dataset size (16, 32 (original), and 64 examples). The results indicate that the average performance across eight zero-shot benchmarks remains remarkably consistent, which highlights the robustness of HC-SMoE with respect to calibration dataset size.
>
> - [Different size of calibration dataset](https://anonymous.4open.science/r/ICML_Rebuttal-0632/Tables/ablation_different_size_of_calibration_dataset.md)
>
> ---
>
> **Q3.** Related Works on EEP
>
> **Response:**
> To provide a clear comparison, we include a table that summarizes the performance and runtime of EEP and HC-SMoE. It merits mention that the reported accuracy for BoolQ and RTE in the EEP paper differs significantly from our results, likely due to differences in evaluation protocols—we use the EleutherAI Language Model Evaluation Harness.
> - [EEP comparison table](https://anonymous.4open.science/r/ICML_Rebuttal-0632/Tables/eep_comparison.md)
> - [Mixtral8x7B-Instruct result](https://anonymous.4open.science/r/ICML_Rebuttal-0632/Tables/experiments_on_mixtral8x7B-Instruct.md)
>
> ---
>
> **Q4.** Fix-Dominant Merging
>
> **Q4.1** What is the definition of "dominant" in Figure 4?
>
> **Response:**
> The definition of “dominant” in fixed-dominant merging refers to a “concept” of dominant expert within a cluster, since it would be the only one expert within the cluster to preserve the original weight vector order. In HC-SMoE, the dominant expert refers to the expert that exhibits the closest proximity to the cluster center.
>
> **Q4.2** Could you explain in more detail how the merging is done?
>
> **Response:**
> Our default fix-dominant merging methodology implements simple average merging, wherein all experts within a cluster contribute equally to the merged representation.
>
> ---
>
> Due to space limitations, we can only provide concise responses here. We have more detailed and comprehensive answers regarding the concerns on section 3.2, section 4.3 and table 4, fix-dominant merging, explanation on table 4 and 5, as well as EEP comparison, which we look forward to discussing thoroughly with the reviewer in the next phase of the review process.

---

> > ### Comment · Reviewer_YKXx · 2025-04-03
> >
> > Thank you for your response! Here are some comments:
> >
> > * I appreciate the additional experiments and hope the authors incorporate them into the revision.
> >
> > * Do we understand why S-prune performs similarly to HC-SMoE on Mixtral8x7B+MATH while being much worse on all other settings?
> >
> > * I still do not fully understand the details of "Fixed Dominant Merging", and some of my other questions are not answered (as you said). I understand that the space is quite limited, and it's impossible to fit all the answers in detail. Please feel free to elaborate on them more in the next response. I will adjust the score accordingly after that.

---

> > > ### Author Response · Authors · 2025-04-06
> > >
> > > Thank you for your reply and engagement! Below we will first address the questions you mentioned in **Rebuttal Comment**, then answer the questions remained in **Rebuttal** section.
> > >
> > > ---
> > >
> > > **Q4. Fix-dominant merge**
> > >
> > > We appreciate the reviewer's interest in our proposed merging method. We acknowledge that Fig. 4 can be further enhanced regarding the merging process, as well as the distinction between expert network weight "features" and expert network output "features."
> > >
> > > Fix-Dominant Merging extends the ZipIt! method [1], which merges neural networks layer by layer through identification of redundancy in their output features. ZipIt first measures output feature similarity at each layer, then merges weight parameters along dimensions corresponding to similar features. This approach enables flexible parameter merging beyond direct one-to-one alignment, thus allowing cross-merging of features based on predefined similarity measures.
> > >
> > > We adapt this concept to MoE expert merging by addressing a key challenge: merging multiple experts within the same cluster rather than merely two networks. To minimize performance degradation, one expert within each cluster retains its original weight ordering and serves as the "dominant expert." We select this dominant expert as the one closest to the cluster center in feature space. During merging, all expert parameters within a cluster undergo averaging with equal weighting. This method provides flexibility in weight feature representation across dimensions. Some dimensions may retain only a single weight feature if dissimilar to all others, while others may preserve multiple weight features based on similarity.
> > >
> > > The algorithm for fix-dominant merging and the updated figure are provided as follows. These additions are intended to clarify the approach and address the reviewer's concerns.
> > >
> > > - [Fix-dom merge algorithm](https://anonymous.4open.science/r/ICML_Rebuttal-0632/algorithm2-fix-dom-merge.png)
> > > - [Fix-dom merge figure](https://anonymous.4open.science/r/ICML_Rebuttal-0632/fix-dom-merge.pdf)
> > >
> > > **Reference**
> > >
> > > [1] Stoica *et.al.* ZipIt! Merging Models from Different Tasks without Training. ICLR 2024.
> > >
> > >
> > > ---
> > >
> > > **Q5.** Do we understand why S-prune performs similarly to HC-SMoE on Mixtral8x7B+MATH while being much worse on all other settings?
> > >
> > > **Response:**
> > >
> > > Thank you for the question. We believe S-prune’s strong performance on MATH stems from both the dataset’s structure and the pruning strategy used.
> > >
> > > Compared to F-prune, S-prune performs notably better on MATH, suggesting that total routing score is a more reliable pruning criterion than activation frequency. MATH inputs often contain LaTeX-like symbols (e.g., '\\', '$'), which can trigger superficial expert activations. S-prune avoids overestimating these by focusing on routing confidence, leading to better pruning.
> > >
> > > While S-prune slightly outperforms HC-SMoE on MATH at 25% pruning (by 0.0038), HC-SMoE surpasses it on 4 out of 8 tasks (e.g., BoolQ, HellaSwag), which require broader expert diversity. In contrast, S-prune excels on tasks like ARC and RTE, which share MATH’s emphasis on formal reasoning.
> > >
> > > Importantly, HC-SMoE is more robust across models, pruning ratios, and calibration data. At 50% pruning, S-prune drops to 0.4192 on MATH, while HC-SMoE retains 0.5861—highlighting the advantage of clustering based on expert output similarity over heuristic usage.
> > >
> > > ---
> > >
> > > **Q6.** Questions on Section 4.3.
> > >
> > > **Response:**
> > > We appreciate the reviewer’s feedback on the variance in Table 4. Hierarchical clustering (HC) is inherently deterministic—given the same data and criteria, it always produces the same result. The observed variance arises from different similarity metrics, not instability in HC itself.
> > >
> > > In our setup, clustering outcomes depend on two factors which shows in Table 4 ablation study: (1) expert representation (e.g., router logits, weights, or averaged outputs), and (2) the linkage criterion used to compute inter-cluster distances. These choices directly affect the final clustering and model performance.
> > >
> > > Our results show that using averaged expert outputs with average linkage offers the best trade-off between effectiveness and stability. Please see Sections 3.2.1 and 3.2.2 for more details.
> > >
> > > ---
> > >
> > > **Q7.** Concerns on Section 1, Section 3.2, Typos, and Table 5.
> > >
> > > **Response:**
> > > We will revise our manuscript to enhance Section 1, Section 3.2 and fix typos accordingly.
> > > - [Updated Table 5](https://anonymous.4open.science/r/ICML_Rebuttal-0632/table5-update.png)
> > >
> > > ---
> > >
> > > We would be glad to further discuss any remaining questions the reviewer may have.

---

### Official Review · Reviewer_A14Y · 2025-03-14

**Overall Recommendation:** 4

**Summary:**

This paper proposes an untrained sparse expert merging strategy, HC-SMoE, which reduces the parameters of  Sparsely activated Mixture of Experts (SMoE) models through expert merging. The clustering strategy adopts hierarchical clustering based on expert output similarity to progressively group experts, while the merging strategy selects frequency-weighted merging to maintain flexibility. Experiments show that HC-SMoE, when applied to Qwen1.5-MoE-A2.7B-Chat and Mixtral 8x7B, achieves an average accuracy drop of 8% and 8.7%, respectively, when reducing the number of experts by half, outperforming existing pruning and merging methods.

## update after rebuttal

Thanks for the authors' feedback, which addresses most of my concerns. I have updated my score accordingly.

**Claims And Evidence:**

The main claims of the paper are generally supported by experiments, but some evidence needs further clarification:
1.	Using expert outputs for hierarchical clustering similarity is claimed to be superior to existing router logit and weight metrics, and experiments on Qwen 45x2.7B validate the effectiveness of output features.
2.	Regarding task independence, experiments are conducted on the C4 dataset and eight zero-shot tasks, along with two domain-specific datasets in the appendix, which partially verify task independence. However, the generalization capability to multimodal domains is not covered, and details on calibration dataset sampling are missing, which may introduce bias.
3.	Regarding the efficiency of MoE models, comparisons with other baselines in terms of memory consumption and computational cost are absent.

**Essential References Not Discussed:**

Model soups: averaging weights of multiple fine-tuned models improves accuracy without increasing inference time

**Experimental Designs Or Analyses:**

In the experimental design, the paper does not compare the memory consumption, computational cost, and other overheads of the SMoE model introduced by other baselines.

**Methods And Evaluation Criteria:**

Methodological soundness: Hierarchical clustering based on expert output similarity is intuitively reasonable, but the rationale for directly adopting Euclidean distance is not discussed.
Evaluation criteria: Using zero-shot tasks and the C4 calibration dataset aligns with the task-independent setting, but the specific sampling scheme for the calibration dataset is not explained.

**Other Comments Or Suggestions:**

It would good to include some discussion of limitations of this work.

**Other Strengths And Weaknesses:**

Strength:
Innovatively introduces hierarchical clustering based on expert output into SMoE expert merging, effectively reducing model memory requirements and computational costs.
Large-scale experiments on Qwen and Mixtral demonstrate practical deployment potential, indicating certain applicability.
Weakness:
Theoretical analysis is still lacking: the theoretical advantages of hierarchical clustering are described vaguely.

**Questions For Authors:**

Q1: Why was Euclidean distance chosen as the distance metric for expert outputs in hierarchical clustering?
Q2: What is the specific sampling scheme for the calibration dataset?

**Relation To Broader Scientific Literature:**

Regarding the field of model merging, the paper only discusses ZipIt but does not address the theoretical connections between HC-SMoE and general model merging techniques (e.g., model soups), making it seem isolated.

**Theoretical Claims:**

Compared to other single-step grouping methods, the paper provides some qualitative analysis of hierarchical clustering, but no detailed theoretical proof is presented in the main text.

---

> ### Author Rebuttal · Authors · 2025-04-01
>
> We appreciate the reviewer’s valuable feedback and effort spent on the review, and would like to respond to the reviewer’s questions as follows.
>
> **Q1.** Theoretical analysis is still lacking: the theoretical advantages of hierarchical clustering are described vaguely.
>
> **Response:**
> Please refer to theoretical justification part in link.
>
> - [Theoretical justification](https://anonymous.4open.science/r/ICML_Rebuttal-0632/Theoretical_Justification.md)
>
> ---
>
>
> **Q2.** However, the generalization capability to multimodal domains is not covered, and details on calibration dataset sampling are missing, which may introduce bias.
>
> **Response:**
> We appreciate the question from the reviewer. This paper concentrates on the MoE-based language domain. Extension to multimodal domains represents an interesting direction for future research but exceeds the current scope of this work. Future investigations could profitably explore the application of HC-SMoE to additional modalities.
>
> ---
>
> **Q3.** Comparison of Memory Consumption, Computational Cost, and Overheads
> We have incorporated a table that compares the runtime and memory usage of HC-SMoE against various baselines. The results demonstrate that HC-SMoE achieves competitive runtime and memory efficiency across different models while maintaining superior performance on benchmarks.
>
> - [Mixtral model experiments](https://anonymous.4open.science/r/ICML_Rebuttal-0632/Tables/ablation_runetime_on_mixtral8x7B.md)
> - [Qwen model experiments](https://anonymous.4open.science/r/ICML_Rebuttal-0632/Tables/ablation_runtime_on_qwen.md)
>
> ---
>
> **Q4.** Choice of Euclidean Distance as Distance Metric for Expert Outputs
>
> **Response:**
> In response to the reviewer’s question about the choice of Euclidean distance as the distance metric for expert outputs in hierarchical clustering, we selected the Euclidean distance due to its effectiveness in measuring the similarity between averaged expert outputs, which are high-dimensional vectors. This choice aligns with prior work such as [3][4], where the Frobenius norm was used to measure pruning error between the original model and the pruned model. In our case, the Euclidean distance is a natural fit, given that the expert outputs are vectors in Euclidean space.
>
> In addition, we show in Table 20 of the paper that even with Euclidean distance, our approach yields high cosine similarity in the final layer output, demonstrating that the choice of Euclidean distance does not hinder performance. We also present empirical results indicating that HC-SMoE produces the best cluster quality, as measured by the silhouette score and Dunn index, when compared to K-means. Please note that in Table 20 the similarity scores between different metrics cannot be directly compared, as the silhouette score and Dunn index are computed on different bases.
>
> ---
>
> **Q5.** What is the specific sampling scheme for the calibration dataset?
>
> **Response:**
> Regarding the calibration dataset, we exactly follow the protocol in [4], using a sampling scheme where we randomly select 32 sentences from the C4 dataset, with each sentence containing 2,048 tokens. The same sampling scheme is used across all experiments to ensure fairness and reproducibility.
>
> ---
>
> **Q6.** Related Work on Model Soup
>
> **Response:**
> We acknowledge the related work on Model Soup and its connection to our approach. Our average merging technique shares similarities with the concept of uniform model soup, where multiple models are averaged to create a unified model. However, while uniform model soup typically involves the combination of multiple models into a single entity, HC-SMoE focuses on merging a set of experts into $r$ clusters, where $r<n$. Our approach exhibits greater complexity, as it incorporates expert clustering based on similarity metrics and a frequency-weighted merging process. We will include citations to the relevant work on uniform model soup in the paper to highlight this connection and differentiate our methodology.
>
> ---
>
> Due to space limitations, we can only provide concise responses here. We have more detailed and comprehensive answers regarding the concerns on limitations discussion of HC-SMoE and other questions, which we look forward to discussing thoroughly with the reviewer in the next phase of the review process.

---

### Decision · Program_Chairs · 2025-05-01

**Decision:**

Accept (poster)

**Comment:**

[1] The main claims of the paper are generally supported by experiments [Reviewer A14Y, cVaE]
[2] This paper proposes a simple and effective task-agnostic method named HC-SMoE to merge the experts in pre-trained mixture-of-expert models. [Reviewer YKXx]
[3] The paper presents a new framework for reducing SMoE model parameters that doesn't require retraining, reduces the parameters of Sparsely activated Mixture of Experts (SMoE) models through expert merging and works across different tasks, making it more generalizable across different datasets. [Reviewer 39Jv]